



# A reassessment of the discrepancies in the annual variation of $\delta$D-H$_2$O in the tropical lower stratosphere between the MIPAS and ACE-FTS satellite data sets

Stefan Lossow[1], Charlotta Högberg[2], Farahnaz Khosrawi[1], Gabriele P. Stiller[1], Ralf Bauer[3], Kaley A. Walker[3], Sylvia Kellmann[1], Andrea Linden[1], Michael Kiefer[1], Norbert Glatthor[1], Thomas von Clarmann[1], Donal P. Murtagh[4], Jörg Steinwagner[5], Thomas Röckmann[6], and Roland Eichinger[7,8]

[1]Karlsruhe Institute of Technology, Institute for Meteorology and Climate Research, Hermann-von-Helmholtz-Platz 1, 76344 Leopoldshafen, Germany.
[2]Stockholm University, Department of Physical Geography, Svante-Arrhenius-väg 8, 10691 Stockholm, Sweden.
[3]University of Toronto, Department of Physics, 60 St. George Street, Toronto, Ontario M5S 1A7, Canada.
[4]Chalmers University of Technology, Department of Earth and Space Sciences, Hörsalsvägen 11, 41296 Göteborg, Sweden.
[5]Max Planck Institute for Extraterrestrial Physics, Giessenbachstraße 1, 85741 Garching, Germany.
[6]Utrecht University, Institute for Marine and Atmospheric Research, Utrecht, Princetonplein 5, 3584 CC Utrecht, The Netherlands.
[7]Deutsches Zentrum für Luft- und Raumfahrt (DLR), Institut für Physik der Atmosphäre, 82234 Weßling, Germany.
[8]Ludwig-Maximilians-University Munich, Meteorological Institute, Theresienstraße 37, 80333 Munich, Germany.

**Correspondence:** Stefan Lossow (stefan.lossow@yahoo.se)

**Version: Thursday 19$^{\text{th}}$ September, 2019  21:22 CET**

**Abstract.** The annual variation of $\delta$D in the tropical lower stratosphere is a critical indicator for the relative importance of different processes contributing to the transport of water vapour through the cold tropical tropopause region into the strato-

sphere. Distinct observational discrepancies of the $\delta$D annual variation were visible in the works of Steinwagner et al. (2010) and Randel et al. (2012), focusing on MIPAS (Michelson Interferometer for Passive Atmospheric Sounding) and ACE-FTS (Atmospheric Chemistry Experiment-Fourier Transform Spectrometer) data, respectively. Here we reassess the discrepancies based on newer MIPAS and ACE-FTS data sets, showing for completeness also results from SMR (Sub-Millimetre Radiometer) observations and a ECHAM/MESSy (European Centre for Medium-Range Weather Forecasts Hamburg/Modular Earth

Submodel System) Atmospheric Chemistry (EMAC) simulation (Eichinger et al., 2015b). Similar to the old analyses, the MIPAS data sets yield a pronounced annual variation (maximum about 75 ‰) while that derived from the ACE-FTS data sets is rather weak (maximum about 25 ‰). While all data sets exhibit the phase progression typical for the tape recorder the annual maximum in the ACE-FTS data set precedes that in the MIPAS data set by 2 to 3 months. We critically consider several possible reasons for the observed discrepancies, focusing primarily on the MIPAS data set. We show that the $\delta$D annual variation in

the MIPAS data is up to an altitude of 40 hPa substantially impacted by a "start altitude effect", i.e. dependency between the lowermost altitude where MIPAS retrievals are possible and retrieved data at higher altitudes. In addition, there is a mismatch in the vertical resolution of the MIPAS HDO and H$_2$O data (being consistently better for HDO), which actually results in an





artificial tape recorder-like signal in $\delta$D. Considering these MIPAS characteristics largely removes any discrepancies between the MIPAS and ACE-FTS data sets and confirms a $\delta$D tape recorder signal with an amplitude of about 25 ‰ in the lowermost stratosphere.

## 1 Introduction

The transport of water vapour from the troposphere to the stratosphere through the cold tropical tropopause layer (TTL) is a critical atmospheric process. On the one hand it directly influences the radiative balance as water vapour is the most important greenhouse gas in this region. On the other hand it constitutes an important contribution to the water vapour budget in the stratosphere. There are multiple transport pathways. The slow ascent within the ascending branch of the Brewer-Dobson circulation (Brewer, 1949) is thought to be the dominating pathway (Fueglistaler et al., 2009). This ascent is accompanied by

large horizontal transport patterns (e.g. Holton and Gettelman, 2001; Bonazzola and Haynes, 2004). They cause a large part of the air trajectories to pass over the Western Pacific, where often the lowest temperatures are encountered and consequently the final dehydration is attained. In addition, the convective lofting of ice particles into the stratosphere is thought to be an important pathway (e.g. Moyer et al., 1996; Dessler et al., 2016). Once the ice particles reach the stratosphere they evaporate and enhance the amount of stratospheric water vapour. Overall, the stratospheric water vapour entry mixing ratios amount

to only $3.5\,\mathrm{ppmv} - 4.0\,\mathrm{ppmv}$ on an annual average (Kley et al., 2000), which is a reduction of three orders of magnitude compared to the mixing ratios commonly observed at the Earth's surface.

The $\delta$D-$H_2O$ (hereafter simply denoted as $\delta$D) isotopic ratio between the minor water vapour isotopologue HD$^{16}$O (hereafter HDO) and the main isotopologue $H_2^{16}O$ (hereafter $H_2O$) is a valuable tool for atmospheric research (e.g. Dansgaard, 1964; Kaye, 1987). This is because it is sensitive to physical, chemical and radiative processes, which can be used to infer the history

of air parcels. As such it can help to discern the relative importance of slow ascent and convective ice lofting for the transport of water vapour into the stratosphere (Moyer et al., 1996). Several parameters are of interest in the tropical lower stratosphere for these investigations, for example the absolute value of $\delta$D, the slope of the HDO–$H_2O$ correlation, any long-term trends in $\delta$D or the regularity and the size of its annual variation. The slow ascent into the stratosphere can to a first order be idealised as a pure Rayleigh fractionation process. Correspondingly a $\delta$D value of about $-900$ ‰ for water entering the stratosphere would

be expected. Observations, however, show typically $\delta$D values between about $-700$ ‰ and $-500$ ‰ (e.g. Moyer et al., 1996; Johnson et al., 2001; Kuang et al., 2003; Webster and Heymsfield, 2003; Nassar et al., 2007; Högberg et al., 2019), clearly indicating an importance of other processes like the convective lofting of ice or the mixing of air masses and supersaturation effects.

Based on MIPAS observations aboard Envisat (Environmental Satellite) Steinwagner et al. (2010) reported a steeper slope

for the HDO–$H_2O$ correlation than expected from Rayleigh fractionation. Discussing different processes influencing the slope they concluded that convectively lofted ice provides the most plausible explanation for the observed steepening of the slope. In their work Steinwagner et al. (2010) only focused on altitudes between $25\,\mathrm{km}$ ($\sim$25 hPa) and $30\,\mathrm{km}$ ($\sim$12 hPa). Below $25\,\mathrm{km}$ ($\sim$25 hPa), they argued, the MIPAS results are affected by a dependency on the lowermost altitude where retrievals are



possible. This lowermost altitude is primarily determined by cloudiness, but also aerosols, increasing water vapour absorption and the atmospheric temperature (Steinwagner et al., 2007; Lossow et al., 2011). The link to results above occurs through error propagation in the MIPAS global fit retrieval approach (von Clarmann et al., 2003). We will refer to this dependency as "start altitude effect" (see more in Sect. 4.2).

An investigation by Dessler et al. (2016) addressed future trends of water vapour in the tropical lower stratosphere, comparing simulations from chemistry climate models and a trajectory model. A substantial increase of about $1\,\mathrm{ppmv}$ until the end of the 21st century was found. About 50% to 80% of this trend could be ascribed to a warming of the TTL in this time period (Dessler et al., 2016). The remainder they attributed to an increase in evaporation of convectively lofted ice. Such a trend would inevitably show up as a trend in the $\delta$D ratio. An observational analysis of such a possible change in the past was performed

by Notholt et al. (2010), employing almost two dozens of balloon-borne observations by the Mark IV interferometer (Toon, 1991) covering the time period between 1991 and 2007 and latitudes from $35°$N to $65°$N. They found a small positive trend in the $\delta$D values of water entering the stratosphere, however this trend was not statistically significant. The authors concluded that, there was no solid indication that the amount of ice entering the stratosphere had changed.

The annual variation of HDO and $H_2O$ in the tropical lower stratosphere is characterised by the atmospheric tape recorder

(Mote et al., 1996; Lossow et al., 2011). The tropical tropopause (cold point) temperature and its annual variation is imprinted in water vapour. The resulting signal is transported upwards and maintained up to about $30\,\mathrm{km}$ ($\sim 12\,\mathrm{hPa}$). For $\delta$D the question arises if there is also such a tape recorder signal found as for HDO and $H_2O$. If not, or if the pattern is obviously disrupted, it would indicate the importance of other processes than slow ascent such as the convective ice lofting. The observational database yields very different pictures to this question. Steinwagner et al. (2010) discussed MIPAS observations and found

a pronounced and coherent tape recorder signal in $\delta$D up to $30\,\mathrm{km}$ ($\sim 12\,\mathrm{hPa}$). Slightly above the tropopause the annual variation had an amplitude of about $60\,\permil$ to $70\,\permil$, which is close to the expected variation based on the annual variation of the tropopause temperature. It can be expected that these results are affected by the start altitude effect described above, as they occur below $25\,\mathrm{km}$ ($\sim 25\,\mathrm{hPa}$). Nonetheless, the tape recorder signal was robust in multiple retrieval setups. Randel et al. (2012) analysed observations of the ACE-FTS instrument aboard SCISAT (Science Satellite) and found a seasonal variation in

the lower stratosphere up to $20\,\mathrm{km}$ ($\sim 56\,\mathrm{hPa}$) and assigned this to the Asian and American monsoon systems. They excluded the possibility that this annual variation is a tape recorder signal. In addition, the amplitude of the annual variation in $\delta$D was considerably smaller than that observed in the MIPAS observations. Recently, Högberg et al. (2019) compared multiple $\delta$D data sets from MIPAS, ACE-FTS as well as the SMR instrument aboard the Odin satellite and concluded that all showed characteristics that typically are associated with the tape recorder. They iterated that pronounced quantitative differences exist,

which would yield different interpretations of the relative importance of convection for lower stratospheric water vapour.

On the modelling side Read et al. (2008) derived an amplitude of about $25\,\permil$ for the annual variation of $\delta$D slightly above the tropopause, using a conceptual two dimensional model incorporating slow ascent, extra-tropical mixing and convection effects. Eichinger et al. (2015b) implemented HDO into the EMAC model and showed a coherent tape recorder signal in $\delta$D up to about $27\,\mathrm{km}$ ($\sim 19\,\mathrm{hPa}$). Slightly above the tropopause the amplitude of the annual variation in $\delta$D amounted to about

$20\,\permil$ (Eichinger et al., 2015b). A sensitivity study showed that the methane oxidation determines the upper end of the tape





recorder signal. Methane is much less depleted in deuterium than water vapour upon the entrance of the stratosphere, i.e. $\delta$D-$CH_4 \approx -80\,‰$ (Röckmann et al., 2011). The water vapour produced in-situ in the lower stratosphere has therefore high $\delta$D values which overshadow the actual tape recorder signal in the EMAC simulation (Eichinger et al., 2015b).

In this work we reassess the differences in the annual variation of $\delta$D in the tropical lower stratosphere between the MIPAS
and ACE-FTS observations. For that we consider newer data sets than employed in the studies of Steinwagner et al. (2010) and Randel et al. (2012), however the discrepancies exist in the same manner. We discuss multiple aspects that could give rise to the observed differences, primarily focusing on the MIPAS data set. A focal point of the discussion are the MIPAS results in the altitude range below $25\,\mathrm{km}$ ($\sim 25\,\mathrm{hPa}$) that have not been included in scientific analyses so far. The paper is structured as follows: In the next section we briefly describe the data sets used in this study and how they have been handled. In Sect. 3
we reassess the observational discrepancies by showing the tropical lower stratospheric $\delta$D time series and by comparing the amplitudes and phases derived for the annual variation. In Sect. 4 we critically consider possible reasons for the discrepancies between the observational data sets. One focus is on the influence that clouds exert on the MIPAS results. In addition, we show that the MIPAS results are influenced by an artefact originating from different vertical resolutions of the retrieved HDO and $H_2O$ data. We demonstrate that the consideration of the MIPAS averaging kernels in the correct way largely removes any
discrepancies between the MIPAS and ACE-FTS data sets and confirm a $\delta$D tape recorder signal with an amplitude of about $25\,‰$ in the lowermost stratosphere. A summary and conclusion of our result will be given in Sect. 5.

## 2   Data sets and handling

In this study we focus on data sets derived from observations of Envisat/MIPAS and SCISAT/ACE-FTS. As noted in the Introduction we here employ newer data sets than used in the original works of Steinwagner et al. (2010) and Randel et al. (2012)
to reassess the observational discrepancies in the annual variation of $\delta$D in the tropical lower stratosphere. As complement we show Odin/SMR results to cover the entire observational database from satellites since the new millennium. In addition, also results from the EMAC simulation (Eichinger et al., 2015b) are considered. Below the data sets are briefly described. For a detailed description of the satellite data sets the reader is referred to the work of Högberg et al. (2019). After the data set description the data handling is explained.

### 2.1   Brief data set description

MIPAS was a cooled high-resolution Fourier transform spectrometer aboard the Envisat satellite (Fischer et al., 2008). Envisat was launched on 1 March 2002 and performed observations until 8 April 2012, when communication with the satellite was lost. Envisat orbited the Earth 14 times a day on a polar, sun-synchronous orbit at an altitude of about $790\,\mathrm{km}$. The Equator-crossing times were 10:00 and 22:00 LT (local time) for the descending and ascending nodes, respectively. MIPAS
measured thermal emission at the atmospheric limb covering all latitudes. In this work we employ results that are based on the version 5 calibration of the European Space Agency and were retrieved with version 20 of the IMK/IAA (Institut für Meteorologie und Klimaforschung" in Karlsruhe, Germany / Instituto de Astrofísica de Andalucía in Granada, Spain) processor





(von Clarmann et al., 2003). The HDO and H$_2$O data are both retrieved from spectral information between 6.7 and 8.0 $\mu$m (1250 – 1483 cm$^{-1}$), for spectral consistency and in an attempt to adjust the vertical resolution of H$_2$O to that of HDO (Steinwagner et al., 2010). Correspondingly, the resulting H$_2$O data set differs from the nominal IMK/IAA H$_2$O data set (applied in the studies of Schieferdecker et al., 2015 or Lossow et al., 2017 for example), which uses more H$_2$O specific microwindows.

Overall, the data set covers the time period from July 2002 to March 2004, which is referred to as the full resolution period of MIPAS.

ACE-FTS is one of three instruments aboard the Canadian SCISAT (or SCISAT-1) satellite (Bernath et al., 2005). SCISAT was launched on 12 August 2003 into a high inclination orbit at an altitude of 650 km. This orbit provides a latitudinal coverage from 85°S to 85°N, but is optimised for observations at high and middle latitudes. ACE-FTS scans the Earth's atmosphere

during 15 sunrises and 15 sunsets a day from about 5 to 150 km altitude. The vertical sampling varies with altitude, ranging from about 1 km in the middle troposphere, to 3 to 4 km at around 20 km, and 6 km in the upper stratosphere and mesosphere. As MIPAS, the ACE-FTS instrument performs observations in the infrared. Here, we use ACE-FTS version 3.5 for the time period from April 2004 to November 2014. HDO information is retrieved from two spectral bands: 3.7 to 4.0 $\mu$m (2493 – 2673 cm$^{-1}$) and 6.6 to 7.2 $\mu$m (1383 – 1511 cm$^{-1}$). The H$_2$O retrieval uses spectral information between 3.3 and 10.7 $\mu$m

(937–2993 cm$^{-1}$).

Odin is a Swedish-led satellite mission that is dedicated to both aeronomy and astronomy observations. Odin was launched on 20 February 2001 into a sun-synchronous orbit with Equator-crossing times of about 06:00 and 18:00 LT on the descending and ascending nodes, respectively. SMR is one of two instrument aboard the Odin satellite and measures the thermal emission at the atmospheric limb (Murtagh et al., 2002). Here, we consider SMR result from the retrieval version 2.1 for the time period

December 2001 to May 2009. The use of later data from this retrieval version is not recommended. HDO (H$_2$O) information is retrieved from an emission line centred at 490.597 GHz (488.491 GHz) as described in the work of Urban et al. (2007).

EMAC (v2.42.0, Jöckel et al., 2010, 2016) is a chemistry-climate model which contains the general circulation model ECHAM (Roeckner et al., 2003) and the MESSy (Jöckel et al., 2005) submodel coupling interface. We use the results of a transient EMAC model simulation with a T42 horizontal ($\sim$ 2.8°x 2.8°) resolution with 90 layers in the vertical and explicitly

resolved middle atmosphere dynamics (T42L90MA). In this setup, the uppermost model layer is centred at around 0.01 hPa and the vertical resolution in the upper troposphere lower stratosphere region is between 500 m and 600 m. To assure that the meteorological situation largely resembles observational data, newtonian relaxation ("nudging") towards data of potential vorticity, divergence, temperature and the logarithmic surface pressure from the Interim ECMWF (European Centre for Medium-Range Weather Forecasts) Reanalysis project (Dee et al., 2011) is applied up to 1 hPa. Besides the standard MESSy

submodels, the H2OISO submodel (Eichinger et al., 2015a) is used in the simulation. It reproduces the EMAC hydrological cycle with additional consideration of HDO and H$_2^{18}$O in all three phases, respectively. For the water isotopologues, equilibrium and kinetic fractionation effects during phase changes in surface fluxes as well as in cloud and convection processes are included (Werner et al., 2011; Eichinger et al., 2015a). In addition to that, Eichinger et al. (2015a) considered H$_2$O and HDO depletion through methane oxidation, which is determined by the mixing ratios of the three oxidation partners Cl, OH, O($^1$D)

and the photolysis rate. The deuterium yield in HD is considered using an empirical approach by McCarthy et al. (2004).



## 2.2 Data handling

In this work we use the "separate" approach for all $\delta$D results as defined by Högberg et al. (2019). That means the results (like the time series) are derived separately for HDO and $H_2O$ and subsequently combined to $\delta$D. The reason for this approach is consistency since the SMR observations of HDO and $H_2O$ are not simultaneous and hence $\delta$D cannot be derived on a single

profile basis. As shown by Högberg et al. (2019) there are some differences between the different approaches to calculate $\delta$D results because of their non-commutativity. However, the main conclusions of this work are not affected by the particular approach chosen. In the Supplement we show a few $\delta$D results for the MIPAS and ACE-FTS data sets based on the "individual" approach, i.e. derived from individual $\delta$D profiles (see Fig. S4).

      The pre-screening of the individual satellite data sets is the same as described in Sect. 2 and 3 of Högberg et al. (2019).

The time series used in this work are based on monthly and zonal averages considering data in the latitude range from 15°S to 15°N. Before the data for a specific bin are averaged an additional screening is performed, as described by Eq. 1 in the Appendix. Subsequently the data are combined. Averages based on less than 20 observations for SMR and MIPAS data sets, respectively, are not considered any further. For the sparser ACE-FTS data sets a minimum of 5 observations is required. In addition, averages that, in absolute values, are smaller than their corresponding standard errors are not further taken into

account.

## 3   Reassessment

In this section the observational discrepancies in the annual variation of $\delta$D in the tropical lower stratosphere between the MIPAS and ACE-FTS data sets are reassessed in two ways. In Fig. 1 the time series of data sets are shown. In Fig. 3 the amplitude and the phase of the annual variation derived from the time series are presented. This comprises not only results

for $\delta$D but also HDO and $H_2O$ for the sake of attribution. In both figures results from the SMR observations and the EMAC simulation are shown as a complement.

      Figure 1 shows the time series of the $\delta$D anomaly with respect to the climatological annual mean for the different data sets. Please note that the time axes vary depending on the data set. For the EMAC simulation arbitrarily the time period from 2000 to 2005 is considered. Other time periods change details but not the overall picture. Gaps in the observational time series are

removed for visual benefits. This concerns primarily the ACE-FTS data sets that have tropical coverage typically only during four months, i.e. February, April, August and October. In 2013 and 2014 there is also some coverage in May and November due to the shifting orbit of SCISAT. This coverage is limited to the very beginning of these months and comprises the northern tropics in May and southern tropics in November. Due to the gap removal the individual monthly averages appear to be valid for longer periods of time in the figure. Also for the SMR data set such gaps occur in early 2005 and 2006. A corresponding

figure with these observational gaps is provided in the Supplement (Fig. S1). In addition, a figure showing only two years of data for the ACE-FTS, SMR and EMAC data sets is supplied (Fig. S2), to provide a more comparable picture to the time coverage of the MIPAS data set.





The $\delta$D results from the different data sets exhibit profound differences. The most coherent temporal behaviour can be observed in the MIPAS data set. It shows a very distinct tape recorder signal in the tropical lower stratosphere, consistent with the results reported by Steinwagner et al. (2010) using an earlier MIPAS retrieval version. Relative to the MIPAS data, the ACE-FTS data set shows a weak annual variation. This is consistent with the results reported by Randel et al. (2012)

for the older retrieval version 2.2, where the annual variation was attributed to the monsoon systems. The annual variation is larger below 50 hPa than higher up. The SMR observations provide information on the $\delta$D variation roughly above 50 hPa. At lower altitudes no $H_2O$ data can be retrieved from its measurements of the 489 GHz band. The variation seems not to be very coherent. During some periods glimpses of a tape recorder signal may be observed, as for example in 2007 and 2008. In contrast, the EMAC simulations show a coherent tape recorder signal which is clearly detectable up to at least 25 hPa. The

annual variation is clearly smaller than in the MIPAS data set, fitting much better that of the ACE-FTS data set.

To characterise the annual variation quantitatively, i.e. its amplitude and phase, we regressed the time series from the individual data sets with a simple regression model that contained an offset, a linear term and the annual variation (see Eq. 2 in the Appendix). The method by von Clarmann et al. (2010) was used to derive the regression coefficients using the standard mean error of the monthly means as statistical weights. Autocorrelation effects and empirical errors as described in Stiller et al.

(2012) are not considered for simplicity. Examplarily Fig. 2 shows the fits (light colours) of the tropical, monthly mean $\delta$D (top panel), HDO (middle panel) and $H_2O$ time series (dark colours) from the different data sets at 70 hPa. In the Supplement a corresponding figure (Fig. S3), focusing on data at 30 hPa, is provided. Figure 3 summarises the derived amplitudes (left panels, see Eq. 3 in the Appendix) and phases (right panels, see Eq. 4 in the Appendix) as function of altitude for all data sets. The top row considers the results for $\delta$D, while the results for HDO and $H_2O$ are shown in the middle and bottom row, respec-

tively. The error bars indicate the $2\sigma$ uncertainty level of the derived quantities. The amplitudes of the annual variation in $\delta$D exhibit clear differences between the MIPAS and ACE-FTS data sets up to about 30 hPa. The data sets from both instruments show the largest amplitudes in the lower stratosphere at about 75 hPa. However, at this altitude also the differences between the MIPAS and ACE-FTS data sets maximise (about 50 ‰). This result is affected by the start altitude effect described in the Introduction. Above, the amplitude decreases significantly. Between 50 hPa and 30 hPa the amplitude for the MIPAS data set

is relatively constant, with slightly more than 20 ‰. Slightly below 40 hPa the ACE-FTS data set shows essentially no annual variation, which may be an artefact. However, this behaviour is also observed for the older version 2.2 data set used in the work of Randel et al. (2012), which in addition covers a shorter time period. Higher up, the amplitude increases again to 10 ‰ close to 30 hPa. The SMR data set exhibits a very large amplitude at its lower boundary. It quickly decreases towards 35 hPa where a good agreement with the ACE-FTS data sets can be observed. Above 30 hPa the amplitudes derived from the individual data

sets are much closer. The EMAC results compare best to the ACE-FTS results below 30 hPa, even though some important differences exist. The simulation exhibits the maximum amplitude at about 90 hPa and does not show a local minimum around 40 hPa as the ACE-FTS observations do. Around 20 hPa the simulation exhibits the best agreement with the ACE-FTS and SMR results.

For all data sets the annual variation exhibits the typical phase progression associated with the tape recorder. Only the ACE-

FTS data set shows a slightly special behaviour between 60 hPa and 40 hPa. In the lower part the phase is relatively constant





while above 45 hPa the behaviour could be interpreted as a phase jump (even though the uncertainties are fairly large). It clearly relates to the low amplitude of the annual variation. Higher up, the phase progression is again as expected for a tape recorder. While the vertical structure of the phase qualitatively agrees, there are substantial quantitative differences among the data sets, similar to the amplitude of the annual variation. Below 60 hPa the difference between the MIPAS and ACE-FTS data

sets typically amounts to 2 – 3 months. The ACE-FTS data set indicates here systematically an earlier occurrence of the annual maximum, comparing better with the EMAC simulation. Also between 35 hPa and 15 hPa there are clear differences between the MIPAS and ACE-FTS data sets. They range between 1 – 3 months, with the latter data set again showing a preceding phase. Above 20 hPa the spread among the data sets is quite large (up to 4 months), however this applies also to the uncertainties of the phase estimates.

The results for the annual variation of HDO qualitatively indicate a good consistency among the data sets. The most obvious quantitative differences between the MIPAS and ACE-FTS data sets occur around 75 hPa, 50 hPa and 20 hPa. The SMR data set exhibits distinctively larger amplitudes below 50 hPa than the other data sets, with few exceptions. Above, the SMR results are close to that of the ACE-FTS and EMAC data sets. The amplitude derived from the EMAC simulation is smallest below 60 hPa. Phase-wise the consistency among the data sets is better for HDO than $\delta$D. Often the differences are within 2 months.

The behaviour that the annual maximum occurs earlier in the ACE-FTS data set than the MIPAS data set persists also for HDO. The EMAC phase precedes all other data sets except above 20 hPa.

The smallest spread among the different data sets in terms of the $H_2O$ annual variation amplitude is observed above 40 hPa. Below 50 hPa the amplitudes derived from the SMR data set are substantially larger than those from the other data sets. This behaviour is the primary reason for the corresponding differences observed in the $\delta$D data. Below 50 hPa there are also

significant differences between the MIPAS and ACE-FTS data sets. The differences amount up to 0.4 ppmv and peak close to 85 hPa. Below 70 hPa the ACE-FTS amplitudes are close to those derived from the EMAC simulation. Higher up, the vertical gradient in the amplitude is smaller in the EMAC simulation leading to clear differences up to about 40 hPa. The comparisons for the phases of the annual variation in $H_2O$ largely resemble the results obtained for HDO.

## 4  Discussion

As shown in the last section, the $\delta$D time series in the tropical lower stratosphere and the derived characteristics of the annual variation exhibit clear differences among the data sets from the MIPAS and ACE-FTS (as well as SMR) satellite observations (and the EMAC simulation). In the following discussion we investigate more thoroughly possible reasons for the differences in the annual variation of $\delta$D in the tropical lower stratosphere, with a clear focus on the MIPAS data set. We will consider the three following aspects:

1. Differences of the MIPAS data set in terms of the temporal sampling of the tropics as well as the inferior vertical resolution relative to ACE-FTS data set.





2. The start altitude effect, i.e. dependency between the lowermost altitude where MIPAS retrievals are possible and retrieved data at higher altitudes, as described in the Introduction.

3. A mismatch in the vertical resolution of the MIPAS HDO and $H_2O$ data that are used in the $\delta D$ calculation.

## 4.1  Temporal sampling and vertical resolution

There are pronounced differences in the temporal sampling of the tropics between the MIPAS and ACE-FTS measurements. While the sun-synchronous orbit of the Envisat satellite yielded a daily coverage of the tropics for MIPAS, the solar occultation measurements of ACE-FTS cover the tropics only at specific times. In general only in February, April, August and October measurements in this geographical region are available (there is some limited observational coverage in May and November in 2013 and 2014). To quantify the impact of this different temporal sampling we subsampled the MIPAS data to the ACE-FTS

coverage. For that we calculated a mean coverage (day of year versus latitude) of the ACE-FTS observations, separately for sunrise and sunset observations. Only MIPAS observations within $\pm 3$ days and $\pm 2°$ of that mean coverage are considered in the subsampled data set. The range in time has been chosen to accommodate the shift of the ACE-FTS coverage over the mission life time as noted in Sect. 3, i.e. a given latitude in the tropics was observed about 3 (6) days later in 2009 (2014) than in 2004. The test results for $\delta D$ are shown in Fig. 4 in light red. For comparison, the MIPAS (red) and ACE-FTS (blue) results from

Fig. 3 are also shown. The results from the subsampled MIPAS data set indicate some systematic effects on the amplitude of the annual variation. Below 60 hPa the subsampled data set yields actually even larger amplitudes, increasing the differences to the ACE-FTS data sets even further. This can be attributed to more obvious differences in HDO than differences in $H_2O$ between the two data sets. Around 55 hPa and between about 25 hPa and 15 hPa the subsampling leads to smaller amplitudes typically corresponding to an improved agreement between the MIPAS and ACE-FTS data sets. In terms of the phase the

subsampling of the MIPAS data results in larger differences below 70 hPa while between 30 hPa and 15 hPa an improvement is observed. Due to the subsampling the error bars of the phase estimates are larger, occasionally leading to an agreement with the ACE-FTS data set in a statistical sense. Overall, the test yields both improvements and deteriorations of the comparison results, clearly indicating that the differences in the temporal sampling between the MIPAS and ACE-FTS data sets are not the main reason for the differences in the annual variation of $\delta D$ in the lowermost stratosphere.

In terms of vertical resolution the data retrieved from the MIPAS observations have a coarser vertical resolution than the ACE-FTS data. In the altitude region of interest the ACE-FTS data exhibit a vertical resolution between 3 km and 4 km. In contrast, the vertical resolution of the MIPAS data used here is between 5 km and 6 km. To emulate the vertical resolution of the MIPAS data the individual ACE-FTS HDO and $H_2O$ profiles were smoothed with a Gaussian window that had a full width at half maximum (FWHM) of 6 km. Subsequently the monthly mean time series were calculated and combined to $\delta D$

according to the "separate" approach employed in this work. After that the regression was performed. The results for this test are shown in light blue in Fig. 4. The amplitude of the annual variation is not really affected, typically there is an agreement with the original results within the error bars. The phases, however, look very different. The characteristics of the tape recorder are essentially lost. This applies not only to $\delta D$ but also HDO and $H_2O$ (not shown). Overall, the test results typically increase the differences between the MIPAS and ACE-FTS data sets. Hence, differences in the vertical resolution between the MIPAS

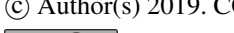



and ACE-FTS data sets can contribute to the observed differences in the annual variation of $\delta$D. We will return to this aspect in Sect. 4.3 where we discuss effects of the vertical resolution mismatch between the MIPAS HDO and $H_2O$ data. In that section the MIPAS averaging kernels are considered which among other data set characteristics also comprise the vertical resolution.

## 4.2 The start altitude effect and its impact

The averaging kernel describes various characteristics of a retrieval, like the altitude where the retrieved information originates from or the vertical resolution. An ideal kernel peaks at the considered retrieval altitude and is symmetric around it. Such exemplary behaviour is however not observed for the MIPAS retrieval of HDO and $H_2O$ in the tropical lowermost stratosphere. The left panel of Fig. 5 shows the displacement between the averaging kernel peak altitudes and the corresponding retrieval altitudes. These results are based on the average over all retrievals in the latitude range between 15°S and 15°N. Positive

(negative) values indicate that the kernels peak at higher (lower) altitudes than the actual retrieval altitudes. Distinct positive displacements are observed below 35 hPa. In this altitude range the displacements are consistently larger for $H_2O$ than HDO. This as also visible in the right panel of Fig. 5 which focuses on the differences between the HDO and $H_2O$ results. At 100 hPa the displacements between the averaging kernel peak altitudes and the corresponding retrieval altitudes amount to 1.25 km for HDO and 2 km for $H_2O$.

A consequence of this behaviour is the start altitude effect, i.e. dependency between the lowermost altitude where MIPAS retrievals are possible and retrieved data at higher altitudes. They are linked through error propagation in the global fit retrieval approach employed for the MIPAS observations. As such, the effect is neither unique to $\delta$D, HDO and $H_2O$ nor the MIPAS retrievals. Steinwagner et al. (2010) identified the start altitude effect as a potential source of concern regarding their slope results for the HDO-$H_2O$ correlation. To avoid related artefacts they considered slope estimates only in the altitude range

between 25 km ($\sim$25 hPa) and 30 km ($\sim$12 hPa). Here, our primary concern is that the start altitude effect has an impact on the annual variation of $\delta$D, HDO and $H_2O$ in the tropical lower stratosphere (below 25 km). There are two aspects to that:

1.  How does the start altitude effect vary during the year?

2.  Does the start altitude itself have an annual variation?

To estimate the start altitude effect and its upper boundary Steinwagner et al. (2010) compared results from two test retrievals:

1.  The standard retrieval, where the spectra of the two lowermost tangent altitudes are (intentionally) discarded (Steinwagner et al., 2007).

2.  A test retrieval, where the spectra of the three lowermost tangent altitudes are omitted to emulate a higher start altitude. Typically the tangent altitudes are separated by 3 km in the MIPAS observations of this altitude region.

The comparisons of the results from two retrievals (following Eqs. 5 to 8 in the Appendix) showed differences in $\delta$D,

$H_2O$ and HDO up to 25 km ($\sim$25 hPa), similar to the differences seen in Fig. 5. Steinwagner et al. (2010) only considered 200 observations (in February, April, May, June and July) for the analysis of the start altitude effect. To get more robust statistics





for all months we expanded the analysis considerably. Overall, more than 53000 observations were employed in the additional test retrieval, corresponding to more than 70% of the tropical observations available in total. Figure 6 shows the differences between the additional test retrieval and the standard retrieval for the individual months.

Considerable differences, i.e. start altitude effects, are observed between the two retrievals. For $\delta$D the differences peak
between $100\,\mathrm{hPa}$ and $70\,\mathrm{hPa}$. In this altitude range the differences vary between $40\,‰$ and $160\,‰$. Higher up, the differences decrease in size. Above $50\,\mathrm{hPa}$ the differences are typically negative. For HDO the differences vary within $\pm0.1\,\mathrm{ppbv}$ at $100\,\mathrm{hPa}$. At $70\,\mathrm{hPa}$ the differences are entirely positive, ranging from $0.02\,\mathrm{ppbv}$ to $0.15\,\mathrm{ppbv}$. Higher up, there is, like for $\delta$D, again a preference for negative differences. For $H_2O$ the largest differences are observed at $100\,\mathrm{hPa}$, ranging from $-1.5\,\mathrm{ppmv}$ to $-0.7\,\mathrm{ppmv}$. Towards higher altitudes the differences become smaller. Above about $40\,\mathrm{hPa}$ they are predominantly positive.
In general, the start altitude effects become insignificant above $30\,\mathrm{hPa}$, which is consistent with the results presented by Steinwagner et al. (2010). There are some differences between results from Steinwagner et al. (2010) and those presented here. This comprises the size of the differences or transition heights between positive and negative values. Overall, the start altitude effect clearly varies as function of month and altitude. Also, the change of the start altitude (given in the legend) differs with month. This makes an influence of the start altitude effect to the annual variation of HDO, $H_2O$ and $\delta$D in the lower
stratosphere rather likely.

The question regarding any annual variation of the start altitude itself is answered by Fig. 7. This figure shows the average start altitude for the individual calendar months considering all available HDO data in the latitude range between $15°S$ and $15°N$. The picture is very similar for the $H_2O$ or $\delta$D data (using the "individual" approach). During boreal winter the average start altitude is located around $100\,\mathrm{hPa}$ or $16.5\,\mathrm{km}$ on the geometric scale. During boreal summer the average start altitude
is lower. Overall, the annual variation in the start altitude (maximum minus minimum) amounts to $36\,\mathrm{hPa}$ in pressure and $1.85\,\mathrm{km}$ in geometric altitude.

An approximation of the impact of the start altitude effect, in combination with the annual variation of the start altitude, on the annual variation of $\delta$D, HDO and $H_2O$ is provided in Fig. 8. These estimates were calculated according to Eq. 12 in the Appendix. However, there are many other possible approaches. The main focus here is on the general distribution of the
impact. There are two key points to be taken from this figure:

1. The temporal variation of the impact does not resemble a tape recorder signal. Accordingly, a correction of the MIPAS data for the impact of the start altitude effect will not universally improve the comparisons to the ACE-FTS data set. At some altitudes the comparison of the annual variation in $\delta$D will improve while at other altitudes they will deteriorate.

2. The size of the impact is substantial, in particular below $40\,\mathrm{hPa}$. Therefore, the start altitude effect has a profound
influence on the amplitude of the $\delta$D annual variation derived from the MIPAS data, clearly requiring some caution.

To conclude the discussion of the start altitude effect and its impact, Fig. 9 presents the characteristics of the annual variation in $\delta$D for MIPAS data with specific start altitudes. The black line is based on data that incorporate all start altitudes, as shown in Fig. 3. There are differences among the results for different start altitudes. In terms of the amplitude of the annual variation they are most pronounced below $50\,\mathrm{hPa}$. The data combining all start altitudes yields amplitudes that are among the largest,



typically only exceeded by the results derived from the data with start altitudes of $17\,\mathrm{km}$ and $18\,\mathrm{km}$. Phase-wise the differences are typically between 1 to 3 months. Below $60\,\mathrm{hPa}$ the data with a start altitude of $12\,\mathrm{km}$ often yield the earliest occurrence of the annual maximum. Higher up, the situation is typically reversed. Here, the data with the largest start altitudes show early occurrences of the annual maximum.

Overall, some of the differences observed in Fig. 9 will have natural causes. Data with a start altitude of $12\,\mathrm{km}$ relate to more or less cloud-free conditions in the TTL while data with a start altitude of $18\,\mathrm{km}$ represent more cloudy conditions in this altitude region. Nonetheless, for all start altitude scenarios the amplitudes derived for $\delta\mathrm{D}$ annual variation in the lowermost stratosphere remain larger than for the ACE-FTS observations. Also, for the phases of the annual variation obvious differences continue to exist. Overall, the start altitude effect clearly impacts the characteristics of the annual variation derived from the
MIPAS data, but obviously more aspects are needed to explain the differences between the MIPAS and ACE-FTS data sets.

### 4.3   Vertical resolution mismatch of the HDO and $H_2O$ data

The comparisons of the annual variation in the lowermost stratosphere (Fig. 3) showed a better agreement between the MIPAS and ACE-FTS data sets for HDO than $H_2O$ in terms of the amplitude. Also, the displacement between the averaging kernel peak altitudes and the corresponding retrieval altitude shown in Fig. 5 is smaller for HDO than for $H_2O$. This warranted a
review of the MIPAS $H_2O$ data set which revealed that its vertical resolution is consistently lower than for HDO in the upper troposphere and lower stratosphere. At $15\,\mathrm{hPa}$ the difference is $0.3\,\mathrm{km}$, at $100\,\mathrm{hPa}$ about $1.4\,\mathrm{km}$ and even more below. This is shown in Fig. 10. The $H_2O$ retrieval has been specifically developed for the joint HDO retrieval (Steinwagner et al., 2007), differing from the nominal $H_2O$ retrieval approach. The goal was to make the vertical resolution of the two species equal, by reducing the vertical resolution of $H_2O$ using a constraint as necessary for a stable retrieval. As such, the resolution mismatch
is only a "residual effect". In contrast, the retrieval of the ACE-FTS observations are unconstrained (Boone et al., 2013), at the expense of not considering effects by the finite field of view of the instrument. No constraints lead by definition to the same vertical resolution for HDO and $H_2O$. This mismatch in the vertical resolution of the MIPAS HDO and $H_2O$ data can lead to spurious results for $\delta\mathrm{D}$ when they are combined, regardless if the combination follows the "separate" or the "individual" approach (Högberg et al., 2019). This statement is of general character for any ratio that is not retrieved directly, not only for
$\delta\mathrm{D}$. In addition, it should be noted that this mismatch is inherently also included in the analysis of the start altitude effect and its impact discussed in the previous section. In the following we quantify the impact of this vertical resolution mismatch and show what a correction means for the comparison of the annual variation in $\delta\mathrm{D}$ between the MIPAS and ACE-FTS data sets.

### 4.3.1   Impact

To illustrate the impact of the vertical resolution mismatch we performed a simple test using the high vertically resolved EMAC
simulation of $H_2O$ (see Sect. 2.1). The tropical monthly means were converted to HDO by assuming a constant $\delta\mathrm{D}$ value of $-575\,‰$ in time and altitude. This constant value roughly corresponds to the overall average value of $\delta\mathrm{D}$ in the tropical lower stratosphere (Nassar et al., 2007; Högberg et al., 2019). Subsequently the HDO data were smoothed in the vertical domain using a Gaussian window with a FWHM of $5\,\mathrm{km}$ to emulate roughly the resolution of the MIPAS HDO data. For $H_2O$ a





window with a FWHM of 6 km was used. From the smoothed data a new $\delta$D distribution was calculated. The deviations from the annual mean for the resulting $\delta$D data are shown in Fig. 11. Due to the mismatch in the vertical resolution of HDO and $H_2O$ the initially time-invariant $\delta$D distribution finally exhibits an artificial tape recorder-like signal. In our simple test this artificial signal has a maximum annual variation of about 14 ‰ close to 80 hPa. Similar results, as described above, were obtained if the

conversion of the simulated $H_2O$ data to HDO assumed an altitude-varying $\delta$D profile (based on ACE-FTS data). Assuming a vertical resolution for $H_2O$ of 7 km instead of 6 km, increases the amplitude of the artificial tape recorder-like signal to values between 15 ‰ and 20 ‰, depending on the HDO conversion approach. Overall, the artificial tape recorder signal derived from these simple tests would only explain a part of the differences in the amplitude of the annual $\delta$D variation observed between the MIPAS and ACE-FTS data sets. In addition, these simple tests show that the resolution mismatch also influences the slope

of the HDO-$H_2O$ correlation in the lower stratosphere. Without the mismatch the HDO-$H_2O$ slope would be less steep. This includes the altitude range between 25 km ($\sim$25 hPa) and 30 km ($\sim$12 hPa) that was analysed by Steinwagner et al. (2010). Accordingly, the actual slope is closer to the slope predicted for the Rayleigh fractionation process, which is associated with the slow ascent of air into the stratosphere by the upwelling branch of the Brewer-Dobson circulation. The importance of non-Rayleigh processes, as convective ice lofting, is consequently smaller.

For a more sophisticated study of the effects caused by the resolution mismatch we involved the actual MIPAS averaging kernels. Unlike in the simple tests above, the kernel shape does not need to have a Gaussian form and as shown in Fig. 5 they do not need to peak at the actual retrieval altitudes. In addition, the averaging kernels account for the start altitude effect. In this more advanced test we considered the EMAC $H_2O$ data for the MIPAS observation period, i.e. from July 2002 to March 2004, again in form of monthly means for the latitude band between 15°S and 15°N. As in the simple tests, these data were converted

to HDO by assuming a constant $\delta$D value of $-575$ ‰. Then, the monthly means of HDO and $H_2O$ were convolved following Eq. 13 in the Appendix with the corresponding averaging kernel and a priori data taken from the MIPAS data set. For a given month, this convolution considered all tropical MIPAS observations. The number of observations ranges from 2100 to 4400, i.e. one monthly mean model profile was convolved a few thousand times. Involving all MIPAS observations aimed to achieve test results as close as possible to the reality observed by MIPAS. For the same reason the individually convolved profiles

were screened according to the standard MIPAS filtering criteria (see Högberg et al., 2019), appropriate to the individual observations. Subsequently all convolved and screened profiles in a given month were combined to a monthly mean. Finally, from the combined monthly HDO and $H_2O$ data the resulting $\delta$D distribution was calculated. The result from this more advanced convolution test is shown in Fig. 12. It exhibits again an artificial tape recorder-like signal as derived from the simple tests. However there are some distinct differences, which most prominently concern the amplitude of the artificial signal. In

September, October and November 2002 the positive deviations amount to around 90 ‰ below 70 hPa, while the negative deviations in March, April and May 2003 range between $-55$ ‰ and $-25$ ‰. This means that the characteristics of the MIPAS retrieval yield much larger artificial tape recorder-like signals as derived from the simple tests, which only focus on the vertical resolution mismatch of the HDO and $H_2O$ data but do not include any other aspects covered by the averaging kernels (as asymmetries or the start altitude effect). In addition to differences in the artefact amplitude, there are also differences in the



phase. Below 70 hPa the extrema occur about one month later in the more advanced convolution test compared to the simple tests shown in Fig. 11. At 50 hPa the delay is about 2 months to 3 months and likewise at 25 hPa.

### 4.3.2 Correction

The artificial tape recorder-like signal derived from the more advanced convolution test can be effectively used to correct the original MIPAS data. This is shown in Fig. 13 where the original MIPAS data set (upper panel, as in Fig 1) was corrected with the test results shown in the previous figure (lower panel). The annual variation of the corrected $\delta D$ data is clearly smaller. At 70 hPa the amplitude amounts to 25 ‰, at 30 hPa to 10 ‰. Both is in agreement with the ACE-FTS data set considering the uncertainties. There are also some phase shifts between the original and corrected time series with respect to the annual variation. This yields both improvements and deteriorations relative to the ACE-FTS data set.

We performed additional advanced convolution tests based on other high vertically resolved simulations of $H_2O$ as well as assuming again an altitude-varying $\delta D$ profile for the conversion of the simulated $H_2O$ data to HDO (see Sect. 4.3.1). For all these tests the correction of the original MIPAS data consistently yields a pronounced reduction of the amplitude of the $\delta D$ annual variation. Below 60 hPa the corrected amplitude amounts to about 25 ‰ on average. Higher up, the variation decreases and above 30 hPa it is less than 5 ‰. The corrected phases exhibit a variation of 1 to 2 months, typically. As for the example above this yields both improvements and deteriorations relative to the ACE-FTS data set.

### 4.3.3 Further considerations

Overall, the vertical resolution mismatch of the MIPAS HDO and $H_2O$ data causes an artificial tape recorder-like signal. Correcting the original MIPAS data for this artificial signal, using the results from the more advanced convolution tests, shows that the MIPAS tape recorder in itself is not entirely artificial. A small signal remains, in good agreement with the ACE-FTS and EMAC data sets. To conclude the discussion on the vertical resolution mismatch and the characteristics of MIPAS averaging kernels we present results from one more advanced convolution test. In this test the EMAC HDO and $H_2O$ data for the MIPAS observation period were taken as they were, i.e. the HDO data were not calculated from $H_2O$ assuming a time-constant $\delta D$ value (see Sect. 4.3.1). On the convolution itself and data handling afterwards nothing was changed. Figure 14 shows correspondingly the characteristics of the annual variation in $\delta D$ derived from the convolved EMAC data (grey). For comparison the results from the original EMAC (black) and MIPAS data (red) are shown. The results highlight that the convolution of the EMAC data with the MIPAS averaging kernels actually yields characteristics for the annual variation that are quite similar to those derived from the original MIPAS data itself. This prominently comprises the larger amplitudes and the phase shift to later occurrences of the annual maximum below 50 hPa. This means that the differences between MIPAS and ACE-FTS data sets can only be understood if the characteristics of the MIPAS retrieval, embedded in the averaging kernels, are taken into account. Since the results for the original EMAC data are in relative good agreement with the ACE-FTS data sets, it can be concluded that also the MIPAS and ACE-FTS data sets are consistent with each other. This echoes the conclusions derived from the correction of the original MIPAS data with the results of the more advanced convolution tests described in Sect. 4.3.2.



## 5   Summary and conclusions

Time series of tropical lower stratospheric $\delta$D exhibit distinct discrepancies. This concerns in particular the characteristics of the annual variation derived from the MIPAS and ACE-FTS data sets, making it difficult to draw robust conclusions relevant for the water vapour transport into the stratosphere. For the amplitude of the annual variation these differences occur prominently

below $30\,\mathrm{hPa}$, with the MIPAS data set consistently indicating larger amplitudes. The maximum difference is observed slightly below $70\,\mathrm{hPa}$ where the amplitude derived from the MIPAS data set is about $50\,‰$ larger than for the ACE-FTS data set (which shows an amplitude of around $25\,‰$). Besides the amplitudes, also the phases of the annual variation exhibit obvious differences between the data sets. In the lowermost stratosphere the differences between the MIPAS and ACE-FTS data sets typically range between 2 months and 3 months, with the annual maximum occurring later in the MIPAS data. Around $40\,\mathrm{hPa}$

a good agreement is observed, but higher up the comparisons deteriorate again. All data sets (including SMR and EMAC), at least, exhibit the typical phase progression associated with the tape recorder. This is contrast to the conclusions of Randel et al. (2012) that attributed the annual variation in the ACE-FTS data to the monsoon systems. In this work we find some atypical phase behaviour for the ACE-FTS data set between $60\,\mathrm{hPa}$ and $40\,\mathrm{hPa}$ which may hint to other influences or a mixture of different signals. However, higher up, the phase progression is again as expected for the tape recorder.

In this work we considered a number of possible reasons for the observational discrepancies between the MIPAS and ACE-FTS data sets. This comprised on the one hand differences in the temporal coverage of the tropics and in the vertical resolution between the two data sets. On the other hand we investigated impacts of special characteristics of the MIPAS retrieval.

Differences in the temporal coverage of the tropics between the two data sets are not a key factor for the observational discrepancies. The MIPAS data have a lower vertical resolution than the ACE-FTS data. Adapting the ACE-FTS data to this

vertical resolution by smoothing them with a Gaussian window (with a FWHM of $6\,\mathrm{km}$) exhibits only little influence on the amplitude of the annual variation of $\delta$D. For the phase this adaption yields an influence, however increasing the differences between the MIPAS and ACE-FTS data sets.

One characteristic of the MIPAS retrieval is a start altitude effect, an inter-correlation between the lowermost altitude where sensible retrievals are possible (typically determined by cloudiness) and results at higher altitudes. This effect arises from

asymmetric kernels and is, as such, not restricted to the data sets considered in this work nor MIPAS results themselves. The start altitude effect exerts an influence on the $\delta$D isotopic ratio and the HDO and $H_2O$ volume mixing ratios up to about $30\,\mathrm{hPa}$ in the MIPAS retrieval. Besides that, the start altitude itself has an annual variation. Both aspects combined result in a profound impact on the amplitude of the $\delta$D annual variation below about $40\,\mathrm{hPa}$. At $100\,\mathrm{hPa}$ a crude estimate indicates a contribution to the amplitude of the annual variation of about $40\,‰$. The temporal variation of the impact is however distinctly different

from a tape recorder signal.

The reassessment of the MIPAS data indicated that some of the discrepancies to the ACE-FTS data set may originate from the $H_2O$ retrieval, which was specifically set up for the joint analysis with HDO (Steinwagner et al., 2007). It was found that there is a remaining mismatch in the vertical resolution of the MIPAS HDO and $H_2O$ data. The resolution of the HDO data is consistently better in the altitude range from $100\,\mathrm{hPa}$ to $10\,\mathrm{hPa}$ by $0.3\,\mathrm{km}$ to $1.4\,\mathrm{km}$. Simple convolution tests with the





vertically high resolved EMAC data, adapted so that there is initially no annual variation in $\delta$D, show that such a mismatch actually causes an artificial tape recorder-like signal in $\delta$D. We expanded these tests by convolving the model data with the actual MIPAS HDO and $H_2O$ averaging kernels and a priori data to obtain realistic results specific to our problem. These tests comprise more MIPAS data set characteristics than just the vertical resolution mismatch, importantly also the start altitude

effect and its impact. Also, these tests exhibited artificial tape recorder-like signals, but even more pronounced. After the correction of the original MIPAS data for the artefact still a tape recorder signal remains, however with a reduced size. On average, the peak amplitude amounts to about 25 ‰ in the lowermost stratosphere, in good agreement with the ACE-FTS and EMAC data sets. In the opposite direction, a convolution of the original EMAC data yields characteristics for the annual variation in $\delta$D that are quite similar to those derived from the original MIPAS data itself. This considers in particular the larger

amplitudes and the direction of the phase shift. Accordingly, the observed differences between the MIPAS and ACE-FTS data sets can be attributed to the characteristics of the MIPAS retrieval (embedded in the averaging kernels) making it essential to consider them.

Finally, both, the simple and more advanced convolution tests show that the vertical resolution mismatch in the MIPAS data artificially enhances the slope of the HDO-$H_2O$ correlation in the lower stratosphere. This concerns also the altitude range

between 25 km ($\sim$25 hPa) and 30 km ($\sim$12 hPa) considered in the work of Steinwagner et al. (2010). Accordingly, the slope is closer to that predicted for the Rayleigh fractionation process and reduces the necessity to involve strong contributions from convectively lofted ice, as was concluded from the original evaluation of the MIPAS dataset (Steinwagner et al., 2010). In a future retrieval version the mismatch in the vertical resolution of the MIPAS HDO and $H_2O$ needs to be fixed.

**Appendix**

Within the Appendix we have collected all equations used in this work.

**Data screening prior the time series calculation**

Before the data for a given month in the tropics were averaged they were screened based on the median and median absolute

difference (MAD, Jones et al., 2012). Data points outside in the interval

$$\langle \text{median}[\boldsymbol{x}(P,t,z)] \pm 7.5 \cdot \text{MAD}[\boldsymbol{x}(P,t,z)] \rangle \qquad (1)$$

were discarded, targeting the most prominent outliers. Here $\boldsymbol{x}(P,t,z) = [x(P,t,z)_1, x(P,t,z)_2, ..., x(P,t,z)_n]$ describes the observations (their total number is $n$) of a given parameter $P$ (i.e. HDO, $H_2O$ or $\delta$D in case of the "individual" approach) at a specific altitude $z$ in the considered month (represented by $t$).

**Calculation of the annual variation characteristics**





To characterise the annual variation, i.e. its amplitude and phase, we regressed the time series from the individual data sets with a simple regression model that contained an offset, a linear term and the annual variation:

$$
\begin{aligned}
f(P,t,z) =\; & C_{\text{offset}}(P,z) + C_{\text{linear}}(P,z) \cdot t + \\
& C_{\text{AO}_1}(P,z) \cdot \sin(2 \cdot \pi \cdot t/p_{\text{AO}}) + \\
& C_{\text{AO}_2}(P,z) \cdot \cos(2 \cdot \pi \cdot t/p_{\text{AO}}).
\end{aligned}
\tag{2}
$$

In the equation $f(P,t,z)$ denotes the fit of time series for a given parameter $P$, the time $t$ (in years) and the altitude level

5   $z$. $C_{\text{offset}}(P,z)$ and $C_{\text{linear}}(P,z)$ are the regression coefficients for the offset and the linear change. A sine and a cosine with a period $p_{\text{AO}}$ of one year were used to parameterise the annual variation. The corresponding regression coefficients are $C_{\text{AO}_1}$ and $C_{\text{AO}_2}$. A more extended regression model, like considering the semi-annual variation, is not meaningful due to the limited tropical coverage of the ACE-FTS data set. Likewise the MIPAS data set is too short to consider any quasi-biennial or ENSO variation. To derive the regression coefficients we followed the method described by von Clarmann et al. (2010). As statistical

10   weights the squared inverse of the standard error of the monthly averages were used. For the sake of simplicity autocorrelation effects and empirical errors as described in Stiller et al. (2012) are not considered. The amplitude $A_{\text{AO}}(P,z)$ of the annual variation is given by:

$$
\begin{aligned}
A_{\text{AO}}(P,z) =\; & \left| \frac{C_{\text{AO}_2}(P,z)}{\sin\{\text{atan}[C_{\text{AO}_2}(P,z)/C_{\text{AO}_1}(P,z)]\}} \right| \\
=\; & |A_{\text{AOsigned}}(P,z)| \\
& \text{for } C_{\text{AO}_1}(P,z) \neq 0,\ C_{\text{AO}_2}(P,z) \neq 0.
\end{aligned}
\tag{3}
$$

We express the phase $P_{\text{AO}}$ by the month in which the annual variation exhibits its annual maximum:

$$
\begin{aligned}
P_{\text{AO}}(P,z) =\; & 1 + s(P,z) \cdot p_{\text{AO}} \cdot 12 - \\
& 12 \cdot \text{atan}\left[ \frac{C_{\text{AO}_2}(P,z)}{C_{\text{AO}_1}(P,z)} \right] \cdot \frac{p_{\text{AO}}}{2 \cdot \pi}.
\end{aligned}
\tag{4}
$$

In this equation $s(P,z)$ is a scaling factor which depends on the sign of $A_{\text{AOsigned}}(P,z)$. If $A_{\text{AOsigned}}(P,z)$ is smaller than zero then $s(P,z)=3/4$, if $A_{\text{AOsigned}}(P,z)$ is larger than zero then $s(P,z)=1/4$. The phase derived by Eq. 4 has a fractional component even though the input data are monthly averages. We kept this fractional component. In correspondence no fraction refers to the beginning of a month while a fraction of 0.5 represents the middle of a month.

**Calculation of the start altitude effect**





The estimation of the start altitude effect was based on the comparison of results from two retrievals, i.e. the standard and a test retrieval, as described in Sect. 4.2. The former retrieval we denote with the subscript "ref", the latter retrieval with the subscript "test". For HDO and $H_2O$ (again represented by $P$) the effect was calculated according to Eq. 5 below:

$$E(P,t,z) = \overline{x_{\text{test}}(P,t,z)} - \overline{x_{\text{ref}}(P,t,z)} \quad \text{with} \tag{5}$$

$$\overline{x_{\text{test}}(P,t,z)} = \frac{1}{n} \cdot \sum_{i=1}^{n} \boldsymbol{x}_{\text{test}}(P,t,z),$$

$$\boldsymbol{x}_{\text{test}}(P,t,z) = [x_{\text{test}}(P,t,z)_1, x_{\text{test}}(P,t,z)_2, ..., x_{\text{test}}(P,t,z)_n], \tag{6}$$

$$\overline{x_{\text{ref}}(P,t,z)} = \frac{1}{n} \cdot \sum_{i=1}^{n} \boldsymbol{x}_{\text{ref}}(P,t,z) \quad \text{and}$$

$$\boldsymbol{x}_{\text{ref}}(P,t,z) = [x_{\text{ref}}(P,t,z)_1, x_{\text{ref}}(P,t,z)_2, ..., x_{\text{ref}}(P,t,z)_n]. \tag{7}$$

Here $\boldsymbol{x}_{\text{test}}(P,t,z)$ and $\boldsymbol{x}_{\text{ref}}(P,t,z)$ describe all results that were available for comparison (i.e. where the start altitude actually changed) in a given month (represented by $t$) and $n$ is their total number. Exactly the same set of observations were considered from the additional test retrieval and the standard retrieval. In addition, data points missing in one data set (due to the MIPAS filtering) were not considered in the other data set and vice versa. For $\delta D$ the start altitude effect was calculated subsequently:

$$E(\delta D,t,z) = \frac{500\,‰}{\text{VSMOW}} \cdot \left[ \frac{\overline{x_{\text{test}}(\text{HDO},t,z)}}{\overline{x_{\text{test}}(\text{H}_2\text{O},t,z)}} - \frac{\overline{x_{\text{ref}}(\text{HDO},t,z)}}{\overline{x_{\text{ref}}(\text{H}_2\text{O},t,z)}} \right] \tag{8}$$

In this equation VSMOW denotes the reference $[D]/[H]$ ratio expressed by the Vienna Standard Mean Ocean Water which has been quantified as $\text{VSMOW} = 155.76 \cdot 10^{-6}$ (Hagemann et al., 1970). In practice the differences (and also the start altitudes) were derived on geometric altitudes, which is the basic vertical coordinate of the MIPAS retrievals. Pressure information simultaneously retrieved from MIPAS observations was carried along in the difference calculation, yielding an average pressure profile for a given month. This profile was subsequently used to interpolate the results onto the pressure grid employed here for presentation.

The change of the start altitude $z_s$ between the two retrievals, corresponding to Eq. 5, was calculated as follows:





$$\Delta z_s(t) = \overline{z_{s\,\text{test}}(t)} - \overline{z_{s\,\text{ref}}(t)} \quad \text{with} \tag{9}$$

$$\overline{z_{s\,\text{test}}(t)} = \frac{1}{n}\sum_{i=1}^{n} \boldsymbol{z}_{s\,\text{test}}(t),$$

$$\boldsymbol{z}_{s\,\text{test}}(t) = [z_{s\,\text{test}}(t)_1, z_{s\,\text{test}}(t)_2, ..., z_{s\,\text{test}}(t)_n], \tag{10}$$

$$\overline{z_{s\,\text{ref}}(t)} = \frac{1}{n}\sum_{i=1}^{n} \boldsymbol{z}_{s\,\text{ref}}(t) \quad \text{and}$$

$$\boldsymbol{z}_{s\,\text{ref}}(t) = [z_{s\,\text{ref}}(t)_1, z_{s\,\text{ref}}(t)_2, ..., z_{s\,\text{ref}}(t)_n] \tag{11}$$

**Calculation of the impact of the start altitude effect**

To provide an estimate of the impact $I$ of start altitude effect, in combination with the annual variation of the start altitude,
on the annual variation of $\delta D$, HDO and $H_2O$ (represented by $P$) we performed the following calculation:

$$I(P,t,z) = \frac{E(P,t,z)}{\Delta z_s(t)} \cdot \left(\overline{z_{s\,\text{ref}}(t)} - \overline{z_0}\right) \quad \text{with} \tag{12}$$

$$\overline{z_0} = \frac{1}{m}\sum_{j=1}^{m} \overline{z_{s\,\text{ref}}(t_j)}.$$

The first part of the equation, i.e. the fraction, relates the changes in $\delta D$, HDO and $H_2O$ between the additional test retrieval
and the standard retrieval to the corresponding changes in the start altitude. As such, this part describes a sensitivity. To es-
timate the impact $I$ this sensitivity was applied to the deviation of the start altitude $\overline{z_{s\,\text{ref}}(t)}$ from the standard retrieval in a
given month to its overall annual mean $\overline{z_0}$ ($\sim 16.0\,\text{km}$, see Fig. 7). $m$ is the total number of months considered, i.e. 12. The
calculation of the impact for $\delta D$ uses $\Delta z_s(t)$ and $\overline{z_{s\,\text{ref}}(t)}$ from the HDO data. The estimated impact as such is a crude estimate.
Different approaches, for example by a different choice for $\overline{z_0}$, will yield different quantitative results for the impact.

**Convolution of higher vertically resolved data**

The convolution of the higher vertically resolved data ($\boldsymbol{x}_{\text{high}}$) onto the vertical resolution of MIPAS ($\boldsymbol{x}_{\text{deg}}$) was achieved by
the equation below which follows the method of Connor et al. (1994):

$$\boldsymbol{x}_{\text{deg}} = \boldsymbol{x}_{\text{a priori}} + \mathbf{A} \cdot (\boldsymbol{x}_{\text{high}} - \boldsymbol{x}_{\text{a priori}}). \tag{13}$$

Here $\mathbf{A}$ and $\boldsymbol{x}_{\text{a priori}}$ describe the averaging kernel and the a priori profile of the MIPAS retrieval.



*Competing interests.* The authors declare that they have no conflict of interests.

*Acknowledgements.* We would like to thank the European Space Agency for making the MIPAS level-1b data set available. The Atmospheric Chemistry Experiment (ACE), also known as SCISAT, is a Canadian-led mission mainly supported by the Canadian Space Agency (CSA) and the Natural Sciences and Engineering Research Council of Canada (NSERC). Odin is a Swedish-led satellite project funded jointly

5    by the Swedish National Space Board (SNSB), the Canadian Space Agency (CSA), the National Technology Agency of Finland (Tekes) and the Centre National d'Etudes Spatiales (CNES) in France. The Swedish Space Corporation has been the industrial prime constructor. Since April 2007 Odin is a third-party mission of ESA. Stefan Lossow was funded by the DFG Research Unit "Stratospheric Change and its Role for Climate Prediction" (SHARP) under contract STI 210/9-2. He thanks the International Meteorological Institute (IMI) hosted by the Department of Meteorology at Stockholm University for financing multiple visits to Stockholm to work personally together with Charlotta

10   Högberg on this topic. Charlotta Högberg thanks the Bolin Centre for Climate Research for financial support to visit the Karlsruhe Institute of Technology to work on this project.

The article processing charges for this open-access publication were covered by a Research Centre of the Helmholtz Association.





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



**Figure 1.** Tropical time series of $\delta$D deviations from the annual mean covering the altitude range between $100\,\text{hPa}$ and $10\,\text{hPa}$. Depending on the data set a different time period is covered, which is indicated in the individual panel titles. For the EMAC simulation the time period from 2000 to 2005 is considered arbitrarily. Observational gaps in the data sets are removed for better visibility, in particular for the ACE-FTS data set. A figure without the gap removal is included in the supplementary material. White areas indicate that no data are available.



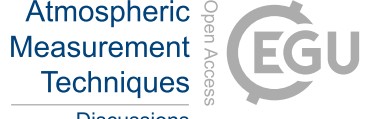

**Figure 2.** Dark colours: Monthly mean time series of $\delta$D (top panel), HDO (middle panel) and H$_2$O (lower panel) at 70 hPa for the different data sets. The error bars represent the standard mean errors. Light colours: The fits of these time series based on the regression model described by Eq. 2.



**Figure 3.** Characteristics of the annual variation for $\delta$D (top row), HDO (centre row) and $H_2O$ (bottom row) in the tropical lower stratosphere as derived from the individual data sets. The left panels show the amplitude and in the right panels the phase is given, expressed by the month where the annual maximum occurs. The error bars represent the $2\sigma$ uncertainty level.



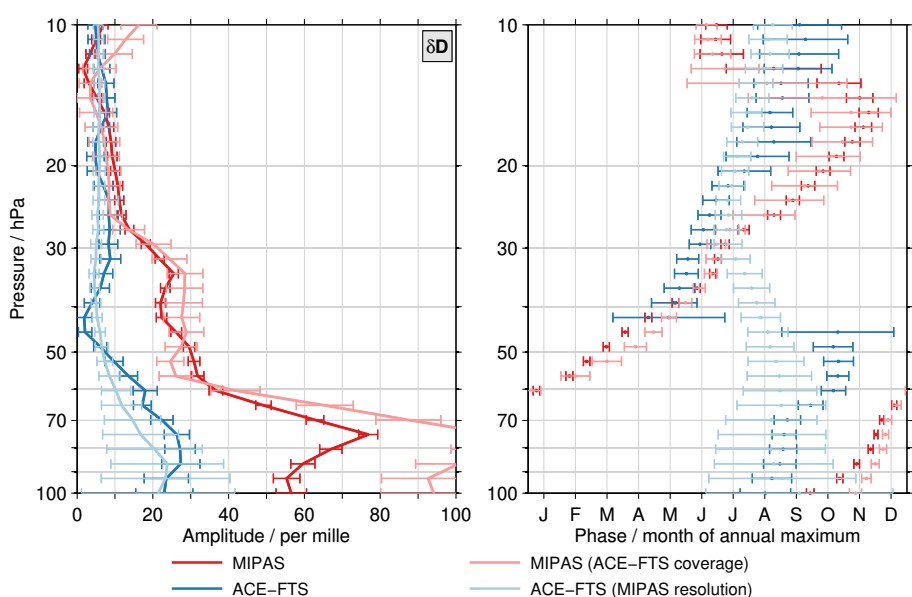

**Figure 4.** As Fig. 3, but here focusing on test results for $\delta$D. Results for a subsampled MIPAS data set that emulates the limited temporal coverage of the ACE-FTS observations in the tropics are shown in light red. The results given in light blue are based on ACE-FTS data whose vertical resolution had been degraded using a Gaussian window with a FWHM of 6 km. The original MIPAS (red) and ACE-FTS (blue) results are shown for comparison.



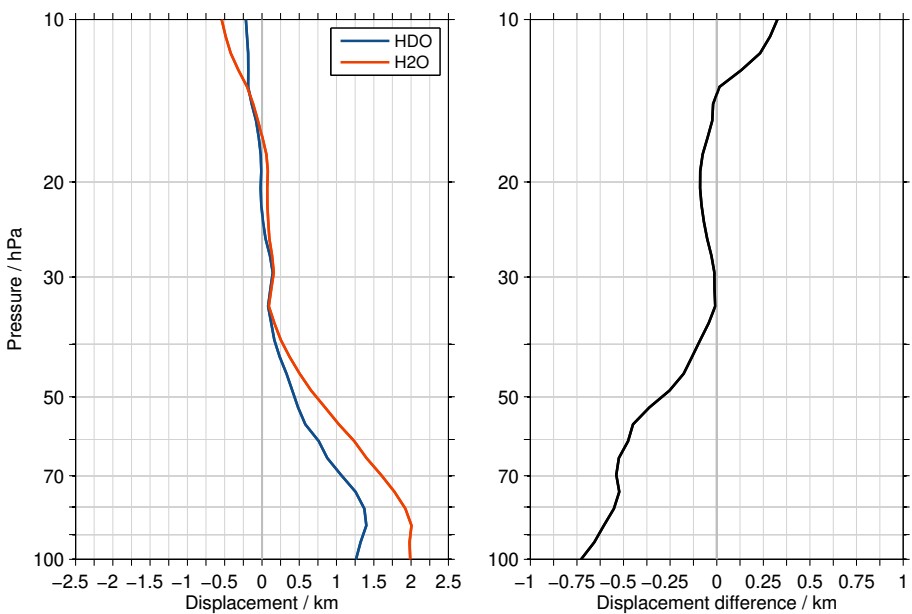

**Figure 5.** Left panel: The displacement between the MIPAS HDO and $H_2O$ averaging kernel peak altitudes and the corresponding retrieval altitudes. Right panel: The differences between the HDO and $H_2O$ displacements.



**Figure 6.** Differences in δD (top panel), HDO (middle panel) and H₂O (bottom panel) between the test and standard retrieval as function of altitude and colour-coded for the calendar months. The calculation of these monthly mean differences follows Eqs. 5 and 8 in the Appendix. In the legend also the change of the start altitude between the two retrievals is indicated for the individual months, derived according to Eq. 9.



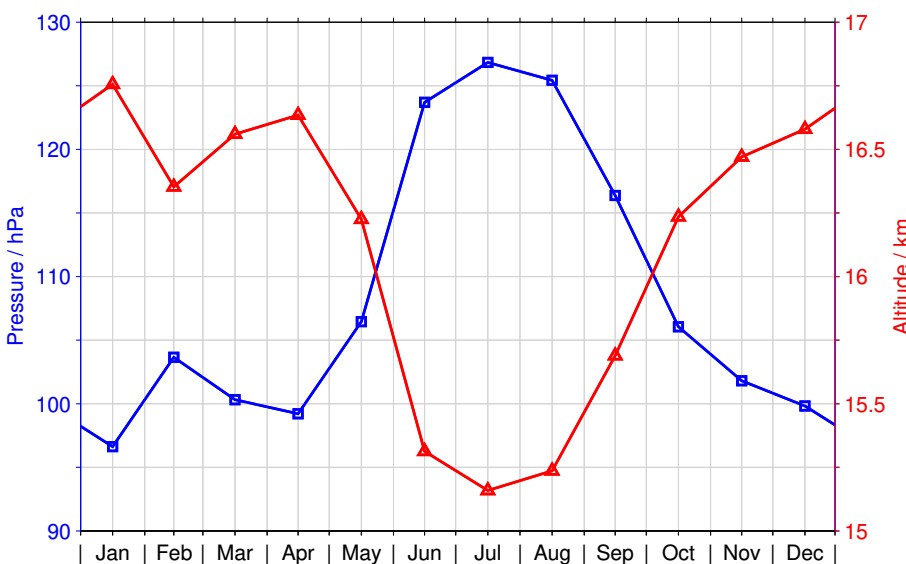

**Figure 7.** The annual variation of the start altitude for the HDO retrieval, i.e. the lowest altitude where sensible data can be derived from the MIPAS observations, considering data in the latitude range between 15°S and 15°N. The blue line shows the variation in pressure (uses the left axis) and the red line considers the geometric altitude (uses the right axis). Note, for better distinction of the two lines the pressure axis is ascending.



**Figure 8.** The approximated impact of the start altitude effect, in combination with with temporal variation of the start altitude, on the annual variation of $\delta$D (top panels), HDO (middle panels) and $H_2O$ (bottom panels) calculated using Eq. 12 in the Appendix.





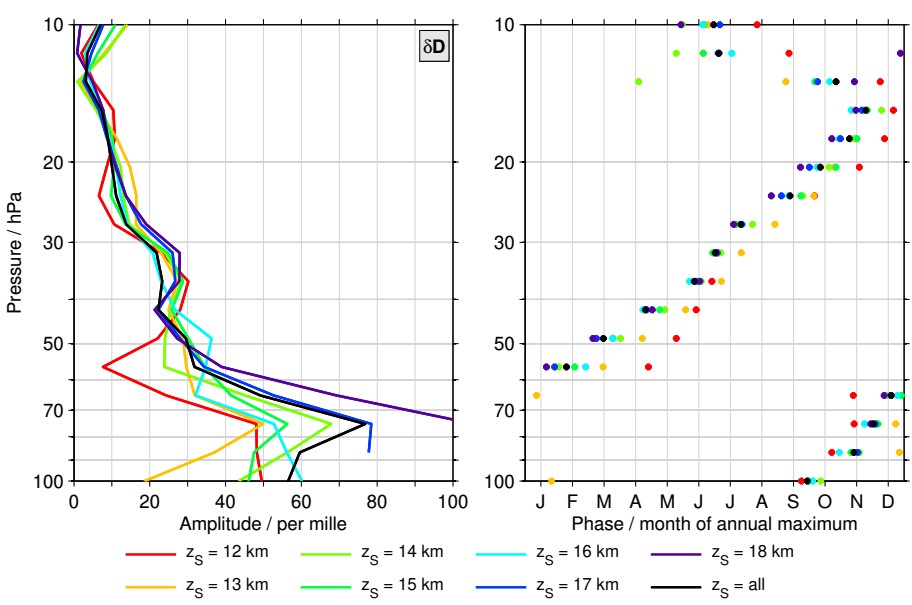

**Figure 9.** Characteristics of the $\delta$D annual variation for MIPAS data with specific start altitudes. The left panels shows the amplitudes while in the right panel the phase estimates are shown. For the sake of simplicity no error bars are provided, unlike in Figs. 3 and 4.



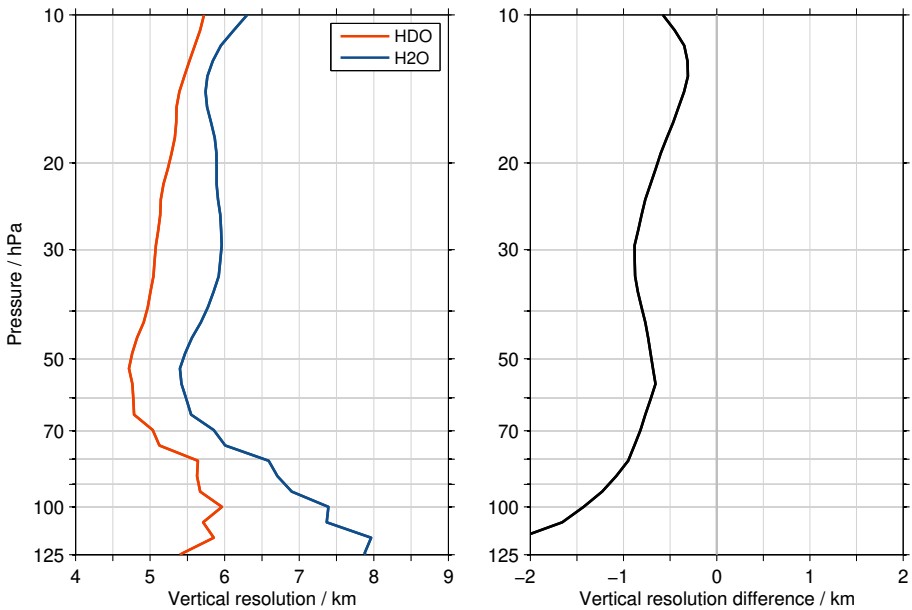

**Figure 10.** The left panel shows the average vertical resolution of HDO (red) and $H_2O$ (blue) for the MIPAS data set, considering all tropical observations. The resolution is derived from the averaging kernel rows using the full width at half maximum. The difference between the resolution of HDO and $H_2O$ is shown in right panel.

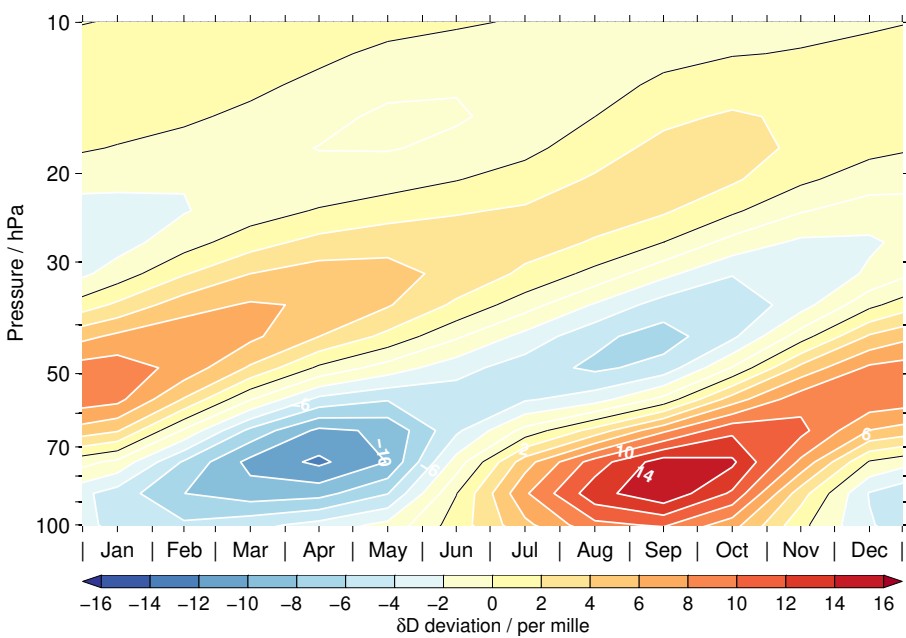

**Figure 11.** An artificial tape recorder-like signal in $\delta$D due to differences in the vertical resolution of the $H_2O$ and HDO data. The results are based on a simple test using the high vertically resolved EMAC data for the latitude band between 15°S and 15°N. See text for more details.



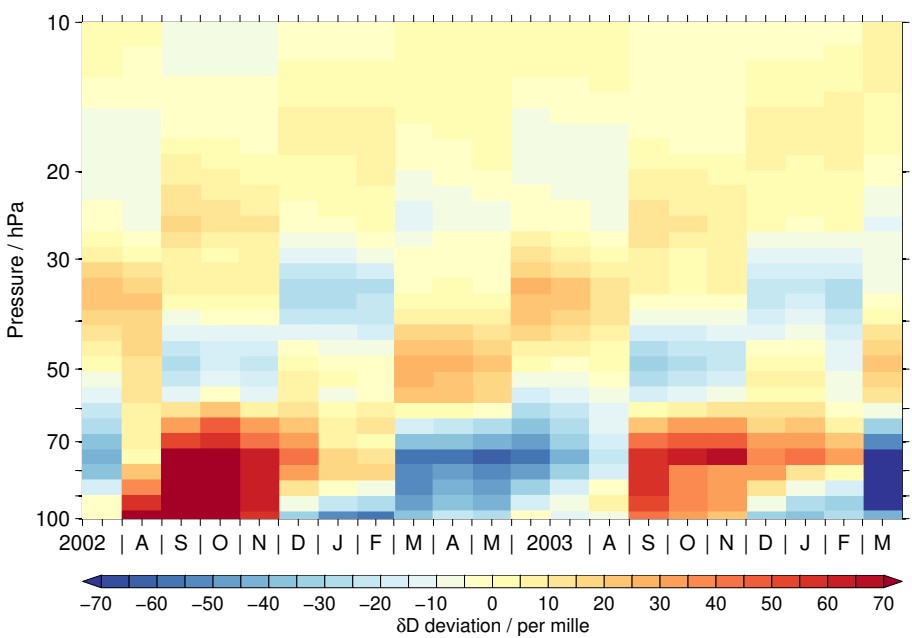

**Figure 12.** An artificial tape recorder-like signal resulting from the convolution of EMAC data with the MIPAS averaging kernels of HDO and H$_2$O. Again, see text for more details.



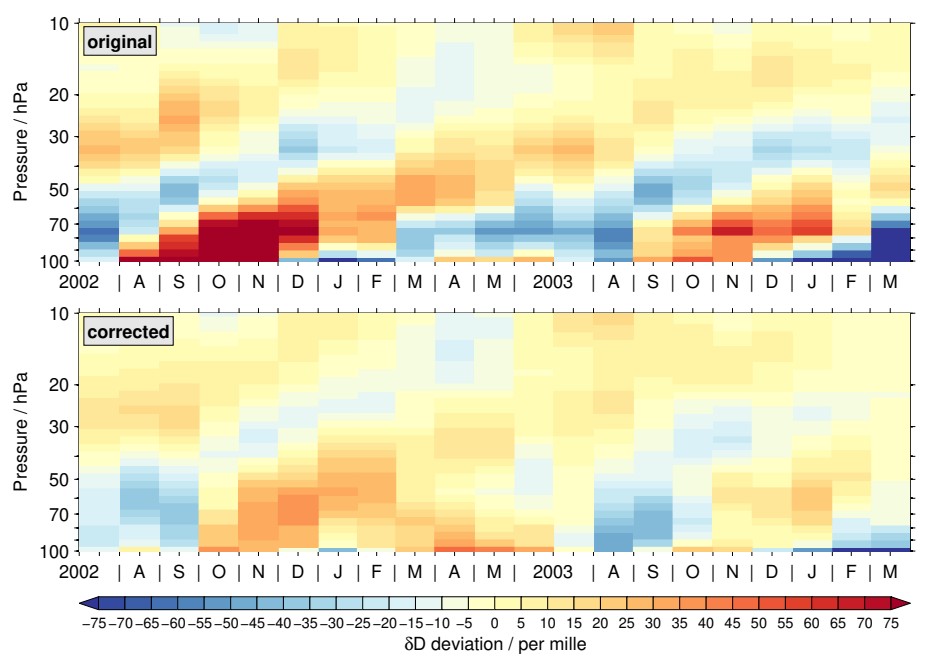

**Figure 13.** Correction of the MIPAS $\delta$D data with the results from the convolution test shown in the previous figure. The top panel shows the original MIPAS time series and the bottom panel the corrected version.

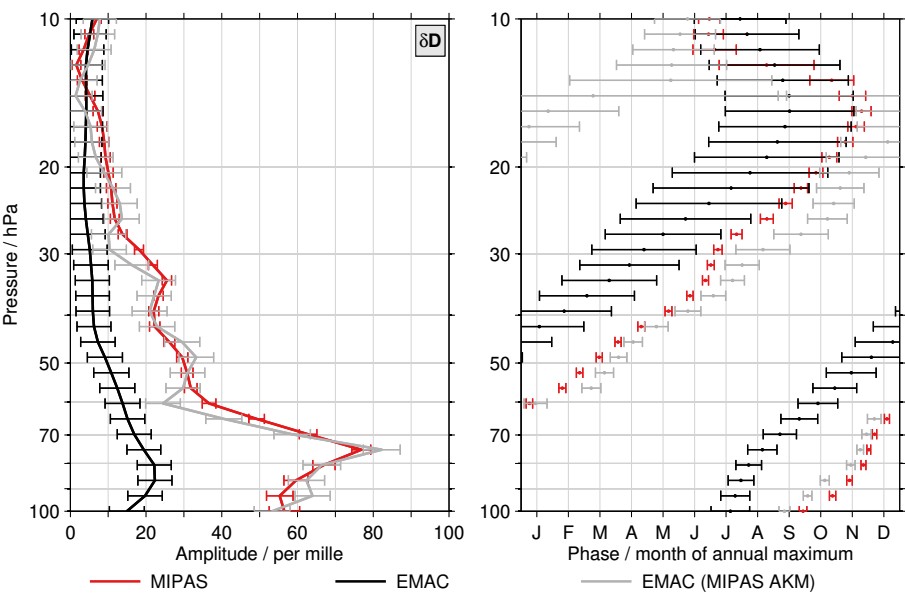

**Figure 14.** As Figs. 3 and 4 but here focusing son EMAC results. For the original EMAC data the results are shown in black, the results based on the convolution with the MIPAS averaging kernels are presented in grey. The original MIPAS results are shown in red.