# Peer review of "A reassessment of the discrepancies in the annual variation of $\delta\text{D-H}_2\text{O}$ in the tropical lower stratosphere between the MIPAS and ACE-FTS satellite data sets"

_Atmospheric Measurement Techniques, 2019_

## Referee Comment (RC1) · Anonymous Referee #2 · 9 Oct 2019

This is an excellent study that shows that the discrepancy between MIPAS and ACE-FTS measurements of the delta-D tape recorder can be explained by the effect of seasonal changes in the lower altitude where MIPAS retrievals are possible. The apparent discrepancy in the delta-D between the two measurements was quite large, and improving our understanding delta-D can help to clarify the contribution of convectively lofted ice to stratospheric water vapor. The study highlights the importance of fully understanding and characterizing the various factors that can affect a satellite retrieval, and shows that such a recharacterization can fundamentally alter the physical

interpretation of the results.

The last sentence of the Abstract does somewhat oversimplify the result. The authors do not show that "MIPAS confirms a delta-D tape recorder signal with an amplitude of about 25 per mille in the lowermost stratosphere." What the authors show (Figure 14) is that when the EMAC simulation (which itself shows a delta-D amplitude of 25 per mille, consistent with the ACE-FTS measurements) is convolved with the MIPAS averaging kernels, then the convolved EMAC simulation gives a result consistent with the MIPAS measurement. A more appropriate phrasing of this entire sentence would therefore be "Considering these MIPAS characteristics largely removes any discrepancies between the MIPAS and ACE-FTS data sets and shows that the MIPAS data is consistent with a delta-D tape recorder signal with an amplitude of about 25 per mille in the lowermost stratosphere."

Figure 7 – I understand that it's easier to see the lines separately with the pressure scale going up, but I really would recommend plotting this with high pressure at the bottom just to avoid confusion.

Page 9 line 22 – "Overall, the test yields both improvements and deteriorations of the comparison results," This is a very awkward phrase. "Overall, the test shows that in some cases agreement improves while in others it becomes worse, . . ." might be better.

Page 12 line 20 - the resolution mismatch is only a "residual effect". I'm not sure what "residual effect" means. I would drop this sentence.

Page 14 line 7 = "Both is" should be "Both are"

---

## Referee Comment (RC2) · Anonymous Referee #1 · 16 Oct 2019

This very well done paper systematically explores the reasons behind the observed differences in \delta-D variation in the lower stratosphere. All of the major contenders are considered including sampling, vertical resolution between instruments and between data products, and the previously known "start altitude effect" in MIPAS. The analysis is convincing and the treatment is thorough, using appropriate averaging kernels and an understanding of the retrievals without becoming overly technical (this is well balanced with supplementary figures in the appendix). It is a fine example of how careful handling of such satellite observations is extremely critical for interpretation. I do

agree with the other reviewer that a rephrasing of the last sentence of the abstract is required. Aside from some very minor suggestions for the authors to consider below, I recommend this paper be published in AMT.

P1, line 15: annual variation in the MIPAS data up to an altitude of 40 hPa is substantially impacted

P4, line 6: A focal point of the discussion is the MIPAS

P8, line 5: ACE-FTS data set systematically indicates an earlier occurrence

Is there qualitative rationale for requiring 20 observations per bin for MIPAS & SMR, and 5 per bin for ACE-FTS?

Consider showing the time series in Fig 1 on the same time axis. As it is, it is hard to interpret even though it is pointed out in the text.

Figure 6: would it be better to show these results as a function of month at a few pressure levels, rather than as profiles? It might be easier to link to the discussion in the text.

---

## Referee Comment (RC3) · Anonymous Referee #3 · 17 Oct 2019

The paper describes the analysis made to understand why ACE-FTS and MIPAS data on the behavior of the deltad-H2O coming from the retrievals of the H2O and HDO volume mixing ratio (VMR) profiles do not agree. In particular MIPAS was finding a tape recorder behavior of deltad with an amplitude larger than ACE-FTS and larger also than what was measured by SMR and predicted by models. It is an interesting investigation and deserves to be published.

However, first of all I fill that the title of the paper and also the paper itself, should clearly state that the MIPAS results discussed here are obtained with the IMK/IAA processor,

because the reported discussion is valid only for the results obtained by that processor and not to MIPAS data itself. In fact, all the analyzed causes are related to features typical of the IMK/IAA analysis method (the starting altitude effect, the different vertical resolution, the Averaging Kernels) and the same results do not apply to retrievals made with different algorithms. This is my main concern. Below find my comments arranged by sections and lines of the discussion paper.

Abstract

Al line 14 it is said that the deltad annual variation is impacted by the start altitude effect. However, in the text (page 12 line 7) it is said that this effect do not removes the discrepancies with ACE-FTS. So, I suggest to change this sentence clearly saying that the start altitude effect alone does not explain the discrepancies among MIPAS and ACE-FTS. Also I would not say in the last sentence that MIPAS confirms the signal amplitude but that MIPAS data are consistent with the ACE-FTS signal amplitude

Introduction

When you introduce the concept of deltad-H2O I think it needs to be explained what deltad stands for.

Line 2 page 3 ' The link to results above' -> 'The link to results at altitudes above'

Line 8 page 3 'The remainder they' -> 'The remainder was'

Line 18-19 page 3 'The observational database yelds very different pictures to this question' -> The reported observations show different answers to this question

Line 6 page 4 'newer data' -> different data (MIPAS data do not change, it is the dataset that has changed)

Line 7 page 4 'however the discrepancies . . ..' -> ' and we find that the same discrepancies exist'

Same line 'aspects that could give rise to' -> 'causes for'

Data sets and handling

Line 19 page 4 'newer' -> 'different'

I have tried to understand the difference between the old and new MIPAS datasets. I could not find any real description of it. Could you please clearly state where the difference is?

Reassessment

In this section Figure 3 is introduced before Figure 2, please check it.

Figure 1 shows the full datasets used in the work. For the sake of comparison I would have liked to have Figure 1 reporting the results on similar time-scale, as it is it is difficult to compare the behavior of deltad for the different instruments. Maybe you can add a figure where the 4 datasets are shown on the same scale (1 year should cover the same length of the x axis) something similar to figure S2 but starting with the same month for all datasets.

Why you blame the start altitude effect on MIPAS and you do not mention the same problem for ACE-FTS? I suppose the two instruments are affected by the cloud coverage in the same way, since they measure in similar spectral regions with the same observing geometry (limb).

Also I think that discrepancies between ACE-FTS and MIPAS could also arise from the fact that MIPAS observes along track (therefore its LOS covers several degrees of latitude) while ACE observes the Sun trough the atmosphere (therefore its LOS covers several degrees of longitude). The horizontal gradients experienced by the two instruments are different, and can cause part of the discrepancies in the results.

Line 24 page 6. I suggest to insert 'Running the model over' before the sentence starting with 'Other time periods'

Line 15 page 7 'Exemplarily' -> As an example

Line 18 page 7 ' as function of' -> as a function of

Line 19 page 7 'row' -> rows

Discussion

I will clearly say in the first paragraph of this section that you are investigating only the possible cause of errors for MIPAS analyses and check if any of them explain the differences between MIPAS and the other datasets.

Page 10 line 6 I do not agree that an ideal kernel is symmetric around its peak for limb observations

In section 4.2 I suppose that the start altitude effect is caused by the use of a fixed vertical (altitude, pressure?) grid in IMK/IAA analysis. I suppose ACE-FTS and SMR use a different strategy. Is it true? The global fit is used at least by both IMK/IAA and ACE-FTS retrievals, so it should affect the results in similar ways.

I have another comment of this section: you test the start altitude effect on real observations. Why don't you use simulated observations where you have all the parameters under control?

In Section 4.3 you say that 'ACE-FTS retrievals are unconstrained at the expenses of not considering effects by the finite field of view' I do not agree with this statement. Unconstrained retrieval does not disregard the field of view effects if they are properly included in the computation of the spectra and the jacobians of the measurements.

---

## Author Comment (AC1) · 6 Nov 2019

**Response to the Comments**
* * *
Colour code:

comments of the reviewer

response by the authors

proposed changes in the manuscript
* * *
General comment:

This very well done paper systematically explores the reasons behind the observed differences in δD variation in the lower stratosphere. All of the major contenders are considered including sampling, vertical resolution between instruments and between data products, and the previously known "start altitude effect" in MIPAS. The analysis is convincing and the treatment is thorough, using appropriate averaging kernels and an understanding of the retrievals without becoming overly technical (this is well balanced with supplementary figures in the appendix). It is a fine example of how careful handling of such satellite observations is extremely critical for interpretation. I do agree with the other reviewer that a rephrasing of the last sentence of the abstract is required. Aside from some very minor suggestions for the authors to consider below, I recommend this paper be published in AMT.

Comment #1:

P1, line 15: annual variation in the MIPAS data up to an altitude of 40 hPa is substantially impacted

Response #1:

The text has been changed accordingly.

Comment #2:

P4, line 6: A focal point of the discussion is the MIPAS

Response #2:

The text was changed to:

The MIPAS results in the altitude range below 25 km (~25 hPa), that have not been included in scientific analyses so far, are considered in particular.

Comment #3:

P8, line 5: ACE-FTS data set systematically indicates an earlier occurrence

Response #3:

The text was changed accordingly.

Comment #4:

Is there qualitative rationale for requiring 20 observations per bin for MIPAS & SMR, and 5 per bin for ACE-FTS?

Response #4:

This choice basically reflects the precision of the data, which is better for the ACE-MIPAS than the MIPAS and SMR data sets. The numbers themselves are empirical, based on working with different water vapour data sets for quite some time.

Comment #5:

Consider showing the time series in Fig 1 on the same time axis. As it is, it is hard to interpret even though it is pointed out in the text.

Response #5:

It is actually unclear to us what was meant with "the same time axis". Using the combined time axis from all data sets would result in a very squeezed representation of the MIPAS data. Alternatively, just showing a single year (maybe just as a climatology) is also difficult to due to limited tropical coverage of the ACE-FTS observations. That is why we picked two years in Fig. S2 in the

Supplement. Overall, our basic idea behind the Fig. 1 (and Fig. 2) was to show the time series that are the basis for the results presented in Fig. 3. The additional figures in the Supplement stemmed from internal discussions, feeling that more on that topic needed to be shown, in line with the comment of the reviewer.

Comment #6:

Figure 6: would it be better to show these results as a function of month at a few pressure levels, rather than as profiles? It might be easier to link to the discussion in the text.

Response #6:

The profiles show the start altitude effect as we defined it by Eqs. 5 and 8. Principally, the effect has been derived by combining retrieval results from a given month. Also, the change of the start altitude (see Eq. 9) depends on the month. To show the data at a given altitude as function of month, the start altitude effect should actually be normalised with the change of the start altitude, otherwise the data are inconsistent from month to month. Admittedly, it never came to our mind to plot the data like that.

---

## Author Comment (AC2) · 6 Nov 2019

**Response to the Comments**
* * *
Colour code:

comments of the reviewer

response by the authors

proposed changes in the manuscript
* * *
General comment:

This is an excellent study that shows that the discrepancy between MIPAS and ACE- FTS measurements of the δD tape recorder can be explained by the effect of seasonal changes in the lower altitude where MIPAS retrievals are possible. The apparent discrepancy in the δD between the two measurements was quite large, and improving our understanding δD can help to clarify the contribution of convectively lofted ice to stratospheric water vapor. The study highlights the importance of fully understanding and characterizing the various factors that can affect a satellite retrieval, and shows that such a recharacterization can fundamentally alter the physical interpretation of the results.

Comment #1:

The last sentence of the Abstract does somewhat oversimplify the result. The authors do not show that "MIPAS confirms a δD tape recorder signal with an amplitude of about 25 per mille in the lowermost stratosphere." What the authors show (Figure 14) is that when the EMAC simulation (which itself shows a δD amplitude of 25 per mille, consistent with the ACE-FTS measurements) is convolved with the MIPAS averaging kernels, then the convolved EMAC simulation gives a result consistent with the MIPAS measurement. A more appropriate phrasing of this entire sentence would therefore be "Considering these MIPAS characteristics largely removes any discrepancies between the MIPAS and ACE-FTS data sets and shows that the MIPAS data is consistent

with a δD tape recorder signal with an amplitude of about 25 per mille in the lowermost stratosphere."

Response #1:

We absolutely agree with that. The text has been changed accordingly.

Comment #2:

Figure 7 – I understand that it's easier to see the lines separately with the pressure scale going up, but I really would recommend plotting this with high pressure at the bottom just to avoid confusion.

Response #2:

The pressure axis is now descending and the corresponding text has been changed.

Comment #3:

Page 9 line 22 – "Overall, the test yields both improvements and deteriorations of the comparison results," This is a very awkward phrase. "Overall, the test shows that in some cases agreement improves while in others it becomes worse, . . ." might be better.

Response #3:

Thanks for the suggestion. It has been included.

Comment #4:

Page 12 line 20 - the resolution mismatch is only a "residual effect". I'm not sure what "residual effect" means. I would drop this sentence.

Response #4:

As already written in our answer to the technical review comments, the change of the $H_2O$ constraint already reduced the differences in the vertical resolution between the H2O and the HDO. In that sense the remaining mismatch is only a residual effect. We have adapted the text as follows:

The $H_2O$ retrieval has been specifically developed for the joint HDO retrieval (Steinwagner et al., 2007), differing from the nominal $H_2O$ retrieval approach. The main reason behind that were actually the differences in vertical resolution between the HDO and $H_2O$ data, with the latter exhibiting a better resolution. To reduce the vertical resolution of the $H_2O$ data the constraint necessary for a stable retrieval was adjusted. This led, overall, to a better agreement of the vertical resolution of the two species. As such the remaining resolution mismatch can be considered as a "residual effect".

Comment #5:

Page 14 line 7 = "Both is" should be "Both are"

Response #5:

Thank you for spotting this.

References:

Steinwagner, J., M. Milz, T. von Clarmann, N. Glatthor, U. Grabowski, M. Höpfner, G. P. Stiller and T. Röckmann, "HDO measurements with MIPAS", *Atmospheric Chemistry & Physics*, 7, 2601 – 2615, doi:10.5194/acp-7-2601-2007, 2007.

---

## Author Comment (AC3) · 6 Nov 2019

**Response to the Comments**
* * *
Colour code:

comments of the reviewer

response by the authors

proposed changes in the manuscript
* * *
General comment:

The paper describes the analysis made to understand why ACE-FTS and MIPAS data on the behavior of the $\delta D$-$H_2O$ coming from the retrievals of the H2O and HDO volume mixing ratio (VMR) profiles do not agree. In particular MIPAS was finding a tape recorder behavior of $\delta D$ with an amplitude larger than ACE-FTS and larger also than what was measured by SMR and predicted by models. It is an interesting investigation and deserves to be published.

However, first of all I fill that the title of the paper and also the paper itself, should clearly state that the MIPAS results discussed here are obtained with the IMK/IAA processor, because the reported discussion is valid only for the results obtained by that processor and not to MIPAS data itself. In fact, all the analyzed causes are related to features typical of the IMK/IAA analysis method (the starting altitude effect, the different vertical resolution, the Averaging Kernels) and the same results do not apply to retrievals made with different algorithms. This is my main concern. Below find my comments arranged by sections and lines of the discussion paper.

General response:

For sure, this work focuses specifically on the MIPAS data set retrieved with the IMK/IAA processor and its characteristics. The start altitude effect itself, is not restricted to the IMK/IAA data set, nor MIPAS.

Overall, we are hesitant to add the processor information to the title for two reasons: (1) It seems an unnecessary complication to add one more abbreviation to the title. (2) We argue, that from the context it should be obvious

which processor is used, given that this is basically a reassessment of the work of Steinwagner et al. (2010) relative to results presented by Randel et al. (2012). In addition, to our knowledge, there are only publications on HDO/δD based on official retrieval products from the IMK/IAA processor (Lossow et al., 2011; Högberg et al., 2019). To our understanding the early work by Payne et al. (2007) was only based on a test data set retrieved with the Oxford processor. Later retrieval versions did not include HDO/δD.

We have additionally noted the IMK/IAA processor at two locations in the revised version of the manuscript: the abstract and the last paragraph of the Introduction that leads into the main part of the manuscript.

**Abstract**

Comment #1:

At line 14 it is said that the δD annual variation is impacted by the start altitude effect. However, in the text (page 12 line 7) it is said that this effect do not removes the discrepancies with ACE-FTS. So, I suggest to change this sentence clearly saying that the start altitude effect alone does not explain the discrepancies among MIPAS and ACE-FTS. Also I would not say in the last sentence that MIPAS confirms the signal amplitude but that MIPAS data are consistent with the ACE-FTS signal amplitude

Response #1:

We have added an additional sentence to make clear that the start altitude effect in itself does not explain the differences between the MIPAS and ACE-FTS data sets.

We show that the δD annual variation in the MIPAS data up to an altitude of 40 hPa is substantially impacted by a "start altitude effect", i.e. dependency between the lowermost altitude where MIPAS retrievals are possible and retrieved data at higher altitudes. In itself this effect does not explain the differences to the ACE-FTS data.

With respect to the last sentence of the abstract, we followed the suggestion of reviewer #2.

Considering these MIPAS characteristics largely removes any discrepancies between the MIPAS and ACE-FTS data sets and shows that the MIPAS data are

consistent with a δD tape recorder signal with an amplitude of about 25 ‰ in the lowermost stratosphere.

**Introduction**

Comment #2:

When you introduce the concept of δD-$H_2O$ I think it needs to be explained what δD stands for.

Response #2:

For completeness we have added now the equation to the Appendix, as done for all other equations.

Comment #3:

Line 2 page 3 "The link to results above" -> "The link to results at altitudes above"

Response #3:

The text has been changed accordingly.

Comment #4:

Line 8 page 3 "The remainder they" -> "The remainder was"

Response #4:

Again, the text has been changed accordingly..

Comment #5:

Line 18-19 page 3 "The observational database yields very different pictures to this question" -> The reported observations show different answers to this question

Response #5:

Thanks for the suggestion! The text has been changed accordingly.

Comment #6:

Line 6 page 4 "newer data" -> different data (MIPAS data do not change, it is the dataset that has changed)

Response #6:

We have not changed anything here, as we actually talk about data sets, not data. We also prefer the word "newer" over "different" because it provides extra information. "Newer" implicitly implies some optimisation of the data sets used in this study with respect to the older data sets employed by Steinwagner et al. (2010) and Randel et al. (2012). Despite these optimisations, the new MIPAS and ACE-FTS δD data sets still exhibit differences in the annual variation.

Comment #7:

Line 7 page 4 "however the discrepancies . . .." -> " and we find that the same discrepancies exist"

Response #7:

The text has been changed accordingly.

Comment #8:

Same line "aspects that could give rise to" -> "causes for"

Response #8:

We kept the text. The word "causes" seems a bit too definitive, given that it is unclear beforehand if these aspects even play a decisive role.

**Data sets and handling**

Comment #9:

Line 19 page 4 "newer" -> "different"

Response #9:

Consistent with response #6 we did not apply any changes here.

Comment #10:

I have tried to understand the difference between the old and new MIPAS datasets. I could not find any real description of it. Could you please clearly state where the difference is?

Response #10:

Given, that the differences in the δD annual variation between the MIPAS and ACE-FTS data sets are independent on the retrieval version we did not include any comparison of the newer data sets used in our work and the old data sets employed in the works of Steinwagner et al. (2010) and Randel et al. (2012). Both the old and the new MIPAS and ACE-FTS data sets are described in detail in the work of Högberg et al. (2019). The MIPAS data set used by Steinwagner et al. (2010) is essentially based on retrieval version 5. The only difference to the retrieval version 20 used here, is the transition from the ESA calibration version 3 to version 5. For the ACE-FTS data sets also the calibration has changed. In addition, there were optimisations with regard to the microwindows considered in the retrieval.

**Reassessment**

Comment #11:

In this section Figure 3 is introduced before Figure 2, please check it.

Response #11:

That is true! We adapted the first paragraph of Sect. 3 as follows to fix this:

In this section the observational discrepancies in the annual variation of δD in the tropical lower stratosphere between the MIPAS and ACE-FTS data sets are reassessed using three figures. Figure 1 shows the time series of the data sets in form of contour plots. In Fig. 2 the focus is on the time series at 70 hPa, not only for δD but also HDO and $H_2O$ for the sake of attribution. In addition, fits from a regression model to quantify the amplitude and the phase of the annual variation (see Eq. 4 in the Appendix) are presented in this figure. Figure 3 shows subsequently the derived amplitudes and phases for the annual variation, again for δD, HDO and H2O. In all three figures results from the SMR observations and the EMAC simulation are shown as a complement.

Comment #12:

Figure 1 shows the full datasets used in the work. For the sake of comparison I would have liked to have Figure 1 reporting the results on similar time-scale, as it is it is difficult to compare the behavior of δD for the different instruments. Maybe you can add a figure where the 4 datasets are shown on the same scale (1 year should cover the same length of the x axis) something similar to figure S2 but starting with the same month for all datasets.

Response #12:

As indicated in response #5 to reviewer #1 our idea was to keep consistency among Figs. 1 to 3 in terms of time coverage. The data presented in Figs. 1 and 2 are the basis for the results shown in Fig. 3. In that sense we are unwilling to change Fig. 1. Evidently, Fig. S2 was our approach to forego any discussion in the direction of showing shorter time periods. However, we are really hesitant to include this in the main part of manuscript. Showing just a single year is difficult to due to limited tropical coverage of the ACE-FTS observations. We improved Fig. S2 so that all data sets except MIPAS start in January and end in December.

Comment #13:

Why you blame the start altitude effect on MIPAS and you do not mention the same problem for ACE-FTS? I suppose the two instruments are affected by the cloud coverage in the same way, since they measure in similar spectral regions with the same observing geometry (limb).

Response #13:

Actually, there are some differences in the cloud influence between the MIPAS and ACE-FTS observations, even though they measure in similar spectral regions. They stem from the fact that MIPAS measures thermal emission at the atmospheric limb while ACE-FTS utilises the solar occultation technique. In terms of the start altitude effect other aspects are of importance. In general, a link between the start altitude and the results at altitudes above can occur through error propagation in a global fit retrieval approach, which is both used for MIPAS and ACE-FTS. A tighter linkage can occur due to retrieval constraints. Also, a retrieval on a fixed grid has the potential to cause a tighter linkage.

These aspects only apply to the MIPAS retrieval which is why we do not put any emphasis on ACE-FTS in this regard.

Comment #14:

Also I think that discrepancies between ACE-FTS and MIPAS could also arise from the fact that MIPAS observes along track (therefore its LOS covers several degrees of latitude) while ACE observes the Sun trough the atmosphere (therefore its LOS covers several degrees of longitude). The horizontal gradients experienced by the two instruments are different, and can cause part of the discrepancies in the results.

Response #14:

It is certainly possible that this aspect has some influence. To provide some rough assessment we performed the following comparison of the characteristics of the annual variation between the MIPAS and ACE-FTS data sets:

MIPAS (15°S- 15°N) minus ACE-FTS (15°S- 15°N) versus

MIPAS (10°S- 10°N) minus ACE-FTS (15°S- 15°N)

Considering MIPAS data only in the latitude range between 10°S and10°N should reduce latitudinal gradients and resulting in a better consistency with the ACE-FTS data set. Overall, the comparison does not yield any obvious differences, indicating that the different latitudinal gradient between the MIPAS and ACE-FTS observation are of secondary importance (in particular to the start altitude effect and the vertical resolution mismatch).

Comment #15:

Line 24 page 6. I suggest to insert "Running the model over" before the sentence starting with "Other time periods"

Response #15:

We have not changed any text with respect to this comment. Actually, the model simulation was run over time period from 1982 to 2010 (Eichinger et al., 2015). We simply picked the time period 2000 to 2005. In that sense we are not sure how to implement "Running the model over".

Comment #16:

Line 15 page 7 "Exemplarily" -> As an example

Response #16:

The text has been changed accordingly.

Comment #17:

Line 18 page 7 "as function of" -> as a function of

Response #17:

Again, the text has been changed accordingly.

Comment #18:

Line 19 page 7 "row" -> rows

Response #18:

Thanks for spotting this! The text has been corrected.

**Discussion**

Comment #19:

I will clearly say in the first paragraph of this section that you are investigating only the possible cause of errors for MIPAS analyses and check if any of them explain the differences between MIPAS and the other datasets.

Response #19:

In the first paragraph of the discussion we state that we clearly focus on the MIPAS data set and list the aspects that we will consider. The term "only the possible cause of errors for MIPAS analyses" feels a bit harsh because differences in the sampling or the vertical resolution between the MIPAS and ACE-FTS data sets are no one's fault. The analysis of the sampling differences is done with the MIPAS data sets, because is the only suitable one. For the differences in the vertical resolution ACE-FTS is used, for the very same reason.

Comment #20:

Page 10 line 6 I do not agree that an ideal kernel is symmetric around its peak for limb observations

Response #20:

We agree that the word "ideal" can be misleading. We have replaced it with "exemplary" which is less strict in describing a desirable model.

An exemplary kernel may peak at the considered retrieval altitude and is symmetric around it. Such behaviour is however not observed for the MIPAS retrieval of HDO and $H_2O$ in the tropical lowermost stratosphere.

Comment #21:

In section 4.2 I suppose that the start altitude effect is caused by the use of a fixed vertical (altitude, pressure?) grid in IMK/IAA analysis. I suppose ACE-FTS and SMR use a different strategy. Is it true? The global fit is used at least by both IMK/IAA and ACE-FTS retrievals, so it should affect the results in similar ways.

Response #21:

As written in response #13 both retrieval constraints and a fixed vertical grid can be of importance. These aspects are not of relevance for ACE-FTS. SMR also uses a global fit retrieval approach, with constraints but a retrieval grid that is aligned to the observed tangent altitudes. In that sense there may be also some start altitude effect in this data set, but we have not investigated that. In general, we were only interested in the overall effect in the MIPAS data set. Investigations of the contributions of different retrieval aspects to this overall effect were not performed.

Comment #22:

I have another comment of this section: you test the start altitude effect on real observations. Why don"t you use simulated observations where you have all the parameters under control?

Response #22:

Not that this is not a good idea, but admittedly it never came to our minds. From the beginning we worked primarily on this specific set of data.

Comment #23:

In Section 4.3 you say that "ACE-FTS retrievals are unconstrained at the expenses of not considering effects by the finite field of view" I do not agree with this statement. Unconstrained retrieval does not disregard the field of view effects if they are properly included in the computation of the spectra and the Jacobians of the measurements.

Response #23:

This is just a characteristic of the ACE-FTS retrieval, not directly related to the unconstrained approach. The half sentence has been removed.

References:

Eichinger, R., Jöckel, P., Brinkop, S., Werner, M., and Lossow, S. (2015). "Simulation of the isotopic composition of stratospheric water vapour – Part 1: Description and evaluation of the EMAC model". *Atmospheric Chemistry & Physics*, 15:5537 – 5555.

Högberg, C., Lossow, S., Khosrawi, F., Bauer, R., Walker, K. A., Eriksson, P., Murtagh, D. P., Stiller, G. P., Steinwagner, J., and Zhang, Q. (2019). "The SPARC water vapour assessment II: profile-to-profile and climatological comparisons of stratospheric δD-H$_2$O observations from satellite". *Atmospheric Chemistry & Physics*, 19:2497 – 2526.

Lossow, S., Steinwagner, J., Urban, J., Dupuy, E., Boone, C. D., Kellmann, S., Linden, A., Kiefer, M., Grabowski, U., Höpfner, M., Glatthor, N., Röckmann, T., Murtagh, D. P., Walker, K. A., Bernath, P. F., von Clarmann, T., and Stiller, G. P. (2011). 2Comparison of HDO measurements from Envisat/MIPAS with observations by Odin/SMR and SCISAT/ACE-FTS". *Atmospheric Measurement Techniques*, 4:1855 – 1874.

Payne, V. H., Noone, D., Dudhia, A., Piccolo, C., and Grainger, R. G. (2007). "Global satellite measurements of HDO and implications for understanding the

transport of water vapour into the stratosphere". *Quarterly Journal of the Royal Meteorological Society*, 133:1459 – 1471.

Randel, W. J., Moyer, E., Park, M., Jensen, E., Bernath, P., Walker, K., and Boone, C. (2012). "Global variations of HDO and HDO/$H_2O$ ratios in the upper troposphere and lower stratosphere derived from ACE-FTS satellite measurements". *Journal of Geophysical Research*, 117(D16):D06303.

Steinwagner, J., Fueglistaler, S., Stiller, G. P., von Clarmann, T., Kiefer, M., Borsboom, P., van Delden, A., and Röckmann, T. (2010). "Tropical dehydration processes constrained by the seasonality of stratospheric deuterated water". *Nature Geoscience*, 3:262 – 266.